# Deepening mechanisms of cut-off lows in the Southern Hemisphere and the role of jet streams: insights from eddy kinetic energy analysis

Henri Rossi Pinheiro[1], Kevin Ivan Hodges[2], Manoel Alonso Gan[3]

[1]Department of Atmospheric Sciences, University of Sao Paulo, 05508-090, Brazil.

[2]Department of Meteorology, University of Reading, Reading, RG6 6UR, United Kingdom.

[3]Center for Weather Forecast and Climate Studies (CPTEC), National Institute for Space Research (INPE), Sao Jose dos Campos, 12227-010, Brazil.

*Correspondence to*: Kevin I. Hodges (k.i.hodges@reading.ac.uk)

**Abstract.** Cut-off lows (COLs) exhibit diverse structures and lifecycles, ranging from confined upper tropospheric systems to deep, multi-level vortex structures. While COL climatologies are well-documented, the mechanisms driving their deepening remain unclear. To bridge this gap, a novel track matching algorithm applied to ERA-Interim reanalysis investigates the vertical extent of Southern Hemisphere COLs. Composite analysis based on structure and eddy kinetic energy budget differentiates four COL categories: shallow, deep, weak, and strong, revealing similarities and disparities. Deep, strong COLs concentrate around Australia and the southwestern Pacific, peaking in autumn and spring, while shallow, weak COLs are more common in summer and closer to the equator. Despite differences, both contrasting types evolve energetically via anticyclonic Rossby wave breaking. The distinct roles of jet streams in affecting COL types are addressed: intense polar front jet correlates with more deep COLs, whereas stronger subtropical jet relates to fewer shallow COLs. The COL deepening typically occurs in the presence of a robust upstream polar front jet which enhances ageostrophic flux convergence and baroclinic processes. The subtropical jet positively correlates with COL intensity, but weakens when considering the seasonality, suggesting uncertainties in this relationship. Additionally, we highlight the significance of diabatic processes in COL deepening, addressing their misrepresentation in reanalysis and emphasizing the need for more observational and modelling studies to refine the energetic framework.

**Keywords**: Cut-off Lows, Vertical Depth, Deepening, Jet Streams, Energetics, Eddy Kinetic Energy.

## 1 Introduction

Cut-off low (COL) pressure systems are closed low-pressure systems that detach or "cut-off" from the main westerly flow (Palmén 1949). These systems manifest as isolated cyclonic potential vorticity anomalies and form both equatorward and poleward of the polar front or mid-latitude jet (Portmann et al. 2021). COLs are known for their slow movement, varying in

duration from short-lived to persisting for several days. Their prolonged periods of precipitation often result in significant accumulations and eventually floods (Singleton and Reason 2007, Llasat et al. 2007, McInnes and Hubbert 2011)

One critical aspect of COLs is their vertical extent, which directly influences their precipitation intensity and duration. Deep COLs, reaching lower atmospheric levels, are usually linked to heavier rainfall compared to shallow systems (Porcù et al. 2007, Pinheiro et al. 2021). Early studies (Palmén 1979) established a groundwork for understanding their vertical structure, revealing their key features such as a quasi-barotropic structure, tropopause folding and thermal dipole patterns. Subsequent studies, including those of Frank (1970) and Porcù et al. (2007), observed a clear relationship between COL vertical extent and the associated cloud and precipitation patterns. More recently, Portmann et al. (2021) highlighted the influence of jet streams on the precipitation intensity of COLs based on their relative position.

Significant progress has been made in understanding the mechanisms driving the development of COLs. Studies have shed light on the crucial role of jet streams in supplying energy to these systems (Pinto and Rocha 2011; Gan and Piva 2013; Ndarana et al. 2021). Their formation often results from a split flow associated with Rossby wave breaking, leading to convergence of ageostrophic geopotential fluxes (hereafter, ageostrophic fluxes) and baroclinic processes. (Ndarana et al. 2021, Pinheiro et al. 2022). Diabatic processes, such as radiative cooling and latent heating, also play a role in COL development (Sakamoto and Takahashi 2005, Garreaud and Fuenzalida 2007, Cavallo and Hakim 2010, Portmann et al. 2018).

Despite these advancements, a comprehensive understanding of COL deepening mechanisms remains unclear. The role of jet streams in deepening COLs and the complex influence of diabatic processes require further investigation. Elucidating these key aspects is crucial for both accurate predictions and improved understanding of COL dynamics. This study aims to address these critical gaps by addressing the following scientific questions:

1.  How sensitive are methods for estimating the vertical depth of COLs?

2.  Can COLs be affectively classified based on their intensity and vertical depth? Do these classifications differ in terms of their spatial distribution and temporal variability?

3.  How do jet streams influence COLs? Does the influence on deepening mechanisms differ between different COL types? Can specific jet stream characteristics be identified as particularly conducive to the deepening of COLs?

By addressing these key questions, we hope to significantly improve our understanding of COLs and their role in atmospheric dynamics. The remainder of the paper is organized as follows. Section 2 outlines the data and methods employed for tracking, estimating vertical depth, and producing composites of structure and energetics of COLs. Section 3 presents the results, and finally, Section 4 provides a summary of the main findings and conclusions.

## 2 Data and methods

### 2.1 Reanalysis dataset and tracking methodology

This study uses the ERA-Interim reanalysis data (Dee et al. 2011) obtained from the European Centre for Medium-Range Weather Forecasts (ECMWF) for the period from 1979 to 2014. ERA-Interim reanalysis employs a spectral model with a N128 reduced Gaussian grid (~80km) and 60 vertical hybrid levels, produced with a four-dimensional variational data assimilation (4D-Var) system.

Before the tracking, the data are spectrally truncated at 42 wavenumbers (T42), and coefficients corresponding to total wavenumbers less than five are set to zero. This is done to reduce noise and eliminate large-scale background influences, similarly to previous work (Pinheiro et al. 2017; 2021; 2022). The established TRACK algorithm (Hodges 1995, 1996, 1999) is employed to track T42 vorticity minima at various pressure levels (1000, 900, 800, 700, 600, 500, 400 and 300 hPa), using a consistent threshold ($-1.0 \times 10^{-5}$ s$^{-1}$) intentionally set relatively low to capture diverse cyclonic systems. The tracking is performed by minimizing a cost-function for track smoothness subject to adaptive constraints on track smoothness and displacement distance. The tracking is the same algorithm as used for extratropical and tropical cyclones (e.g., Hodges et al. 2011, 2017), but with some adjustments to the slower movement of COLs, as discussed in Pinheiro et al. (2019). The resulting tracks are filtered based on horizontal wind components (u, v) at four 5$^{\circ}$ geodesic offset points from the vorticity center: 0$^{\circ}$ (u>0), 90$^{\circ}$ (v<0), 180$^{\circ}$ (u<0) and 270$^{\circ}$ (v>0) relative to north. This retains only closed cyclonic centers, as discussed in Pinheiro et al. (2019). The retained tracks are those that either move equatorward and reach latitudes north of 40$^{\circ}$S or originate north of 40$^{\circ}$S and persist for at least 24 hours.

Spatial statistics of the COLs are computed using the track information and spherical kernel estimators for track density and mean intensity (Hodges 1996). Track density represents COLs per season per unit area (5$^{\circ}$ spherical cap $\cong 10^6$ km$^2$), while mean intensity derives from T42 relative vorticity, scaled by -1 for the Southern Hemisphere.

### 2.2 Approach to estimate the vertical depth of COLs

To analyze and quantify COLs, numerous algorithms have emerged, though only a few of them address multi-level cyclone detection and their connections. Existing algorithms establish cyclone position correspondence between levels based on feature point distances, utilizing geopotential or vorticity centers. These methods span from basic search tasks (Lim and Simmonds 2007, Porcù et al. 2007) to more advanced techniques based on optimal solution (Lakkis et al. 2019).

In this study, the track matching algorithm introduced by Hodges et al. (2003) and used to match tracks in different datasets (Bengtsson et al. 2009, Hodges et at. 2011, 2017, Pinheiro et al. 2020) is employed to match tracks between different pressure levels. The algorithm is used to compare tracks across different pressure levels by defining a mean separation distance ($d_m$), chosen here to be 5 degrees geodesic, and considering overlaps in time ($\chi$) between corresponding points in the tracks.

Temporal overlaps between tracks will be determined following a sensitivity analysis, as discussed in Section 3.1. The percentage of points overlapping in time is calculated using the approach described in Hodges et al. (2003), defined as follows:

$$\chi = 100[2n_m/(n_1 + n_2)]$$

where $n_m$ is the number of points that match in time, and $n_1$ and $n_2$ are the number of points in the track corresponding to different pressure levels.

Since our focus is on the upper-level forcing driving surface cyclone development and the mechanisms governing this interaction, our method works top-down, starting with two levels at a time and progressively extending the matches to adjacent pressure levels. Vorticity tracks ($\xi_{300}$) are matched against $\xi_{400}$ if the mean separation distance is ≤5º and temporal overlap exists. Successful matches indicate extending COLs; unmatched $\xi_{300}$ tracks imply the COL is confined to 300 hPa. An iterative process continues to lower levels (e.g., 500 hPa, 600 hPa, and so on), stopping at the last match at 1000 hPa. The deepest successful match determines the COL vertical extent.

Complex interactions between COLs and lower-level cyclonic features makes capturing the full range of coupling processes challenging. Our algorithm, while simpler than the optimal solution-based approach in Lakkis et al. (2019), consistently establishes vertical associations and a sequential stacking process. However, diverse matching procedures might result in distinct COL evolution outcomes, although prior research has shown similar results between bottom-to-top and top-to-bottom approaches (Lakkis et al. 2019).

**2.3 Compositing analysis of the structure and energetics of COLs**

Employing a system-centered compositing method similar to that of Pinheiro et al. (2022), we investigate the composite structure of COLs and their eddy kinetic energy (EKE) budget. Initially, COLs are identified and categorized into shallow, medium and deep types based on their vertical extent, however, due to space constraints only the findings concerning deep and shallow COLs are presented in the paper. Shallow COLs are limited to the upper troposphere, extending no lower than 400 hPa, while deep COLs originate at high levels and extend down to 800 hPa or lower. The depth levels are chosen to capture the contrasting vertical extents of COLs, which are typically found either in the upper or lower troposphere. These depth categories, each accounting around 30% of the total COLs, are chosen to ensure a balanced representation across different types in our analysis. Additionally, we classify COLs as strong (above the 50th percentile) or weak (below the 50th percentile) according to their maximum intensity observed along each track, based on the 300-hPa vorticity.

Atmospheric fields and energetic quantities are sampled on a 25° latitude-longitude rectangular grid, initially defined centered on the equator then rotated to the COL 300hPa vorticity center, with a horizontal resolution of 0.5°. The sampling is performed at each vertical level from 1000 hPa to 100 hPa, taking the 300-hPa track point as the reference. We present vertical composites for west-east cross-sections centered on the vorticity center. These composites are produced for various time intervals relative

to the peak intensity of COLs, but only for times within $\pm$ 48 hours of the peak intensity are shown. Extending the compositing window beyond this timeframe often introduces noise due to variations in COL lifetimes.

Using the Orlanski and Katzfey (1991) method, we investigate the energy dynamics associated with the COL deepening through the EKE budget. This approach considers essential mechanisms such as baroclinic and barotropic conversions and ageostrophic flux convergence (downstream development). In this study, the focus is on baroclinic conversion and ageostrophic flux convergence, the two primary EKE budget mechanisms in COLs (Gan and Piva 2016, Ndarana et al. 2021, Pinheiro et al. 2022).

The budget residual accounts for processes not fully captured by the EKE budget calculation, including friction and discretization errors such as interpolation and finite differences. Errors could also arise from how diabatic processes are represented in reanalysis products (Pinheiro et al. 2022). Time-mean quantities are calculated for each month averaged over 28–31 days for the 6-hourly data, separately for each individual month and year.

**2.4 Link with jet streams**

The connection between COL extent and jet streams is also explored using Pearson correlation coefficients and scatter plots. Following Bals-Elshol et al. (2001), the characteristics of the polar front and subtropical jets are defined by averaging the 300-hPa zonal mean zonal wind at 50°S-65°S and 25°S-35°S, respectively. The polar front jet, characterized by large zonal variations, is defined as a wide latitudinal band, contrasting with the subtropical jet which exhibits relatively minor spatial and seasonal variations in its position within the Southern Hemisphere (Simmons and Jones 1998). Correlation coefficients are employed to verify the relationship between jet strengths and COLs, with significance exceeding 95% and 99%.

**3 Results and discussion**

**3.1 Sensitivity of track matching algorithm for the estimation of COL depth**

To assess the sensitivity of the track matching algorithm in determining COL depth, we investigate the influence of the mean separation distance ($d_m$) and temporal overlap ($\chi$) between tracks at different pressure levels. We systematically varied $\chi$ from 1% to 100% while keeping $d_m$ fixed at 5° geodesic. This choice is based on observations that the depth estimation is more sensitive to $\chi$ than $d_m$. Setting $d_m$ to a large value leads to significant rise in the number of matches; however, this can introduce false matches involving unrelated cyclonic systems. Therefore, we set $d_m$ to 5 degrees geodesic which is suitable considering the typical small vertical tilt of COLs (refer to Fig. 9 in Pinheiro et al. 2021).

A sensitivity analysis varied $\chi$ across a range of values. Setting $\chi$ to 1% requires a minimum 1% time overlap between tracks at different pressure levels. Table A.1 in the Appendix shows that 20.3% of $\xi_{300}$ COLs extend to the surface in at least one time step, indicating interconnected cyclonic features across all levels from 300 hPa to 1000 hPa. The number of systems reaching the surface remains relatively consistent for $\chi$ values ranging from 1% to 25% at 20.3% to 19.5% of all COLs. However, matches significantly decrease when $\chi$ exceeds 50%, likely due to COLs only associating briefly with lower-level

features. With χ set to 75%, less than 10% of systems are identified as deep COLs, suggesting an underestimation due to association with short-lived lower-level features. Relaxing the χ threshold increases the chance of capturing stacked cyclonic system, guaranteeing vertically aligned or tilted COLs across adjacent levels using $\chi$=1% and $d_m$=5°.

Using geopotential data instead of relative vorticity to estimate COL depth is an alternative approach. However, some care is required in selecting an appropriate threshold as geopotential magnitude generally increases with height in baroclinic systems.
Previous studies (Porcù et al. 2007, Barnes et al. 2021) used varying geopotential thresholds for each pressure level, requiring different subjective thresholds at each level. In contrast, vorticity-based identification does not require vertical adjustments, as vorticity measures atmospheric flow rotation. Moreover, weaker COLs are less likely to be identified in a geopotential field-based tracking method (Pinheiro et al. 2019).

### 3.2 Relationship between intensity and vertical depth of COLs

Classifying COLs by grouping similar systems is crucial for uncovering the key factors influencing their diverse types, allowing a deeper understanding of their dynamics and providing a framework for evaluating climate impacts. This section explores the contrasting characteristics among four COL categories: shallow, deep, weak, and strong COLs (Figure 1). These categories are defined based on intensity and vertical extent, as detailed in Section 2.3. While intermediate-level COLs exist, this study focuses on the contrasting features of their vertical extent. This analysis does not focus on vertical tilt, which is
comprehensively addressed in Pinheiro et al. (2021).

Figure 1 shows the annual track density of COLs in the four categories described above. Deep and strong systems exhibit similar distribution patterns, both primarily concentrated in Australia and the southwestern Pacific, with a secondary maximum in the southeast Pacific off the west coast of South America. These two groups exhibit similar intensity patterns, with maxima (blue dots) located in Australia and upstream of the main continents, as previously demonstrated (Barnes et al. 2021, Pinheiro
et al. 2021, 2022). Shallow and weak COLs also share similarities; they are less intense and more dispersed than the deep and strong systems. These systems are predominantly found in the South Atlantic, southeast Africa (Madagascar), and the South Indian Ocean, and they are more frequently found equatorward than their deep and strong counterparts, a notable feature over the central-eastern Pacific Ocean.

By contrasting the tracks within each category, we found that 81% of deep COLs correspond with the 50th percentile of
strongest systems, suggesting a significant correlation between the two categories. However, it is noteworthy that 19% of deep COLs fall into the category of weak COLs, indicating a level of variance within this classification. Similarly, 71% of weak COLs correspond to shallow COLs. Despite the similarities shared between strong and deep COLs, as well as weak and shallow COLs, it is important to recognize that they are not entirely comparable. These distinctions arise from differences in classification and the dynamics inherent in these systems and their development processes.

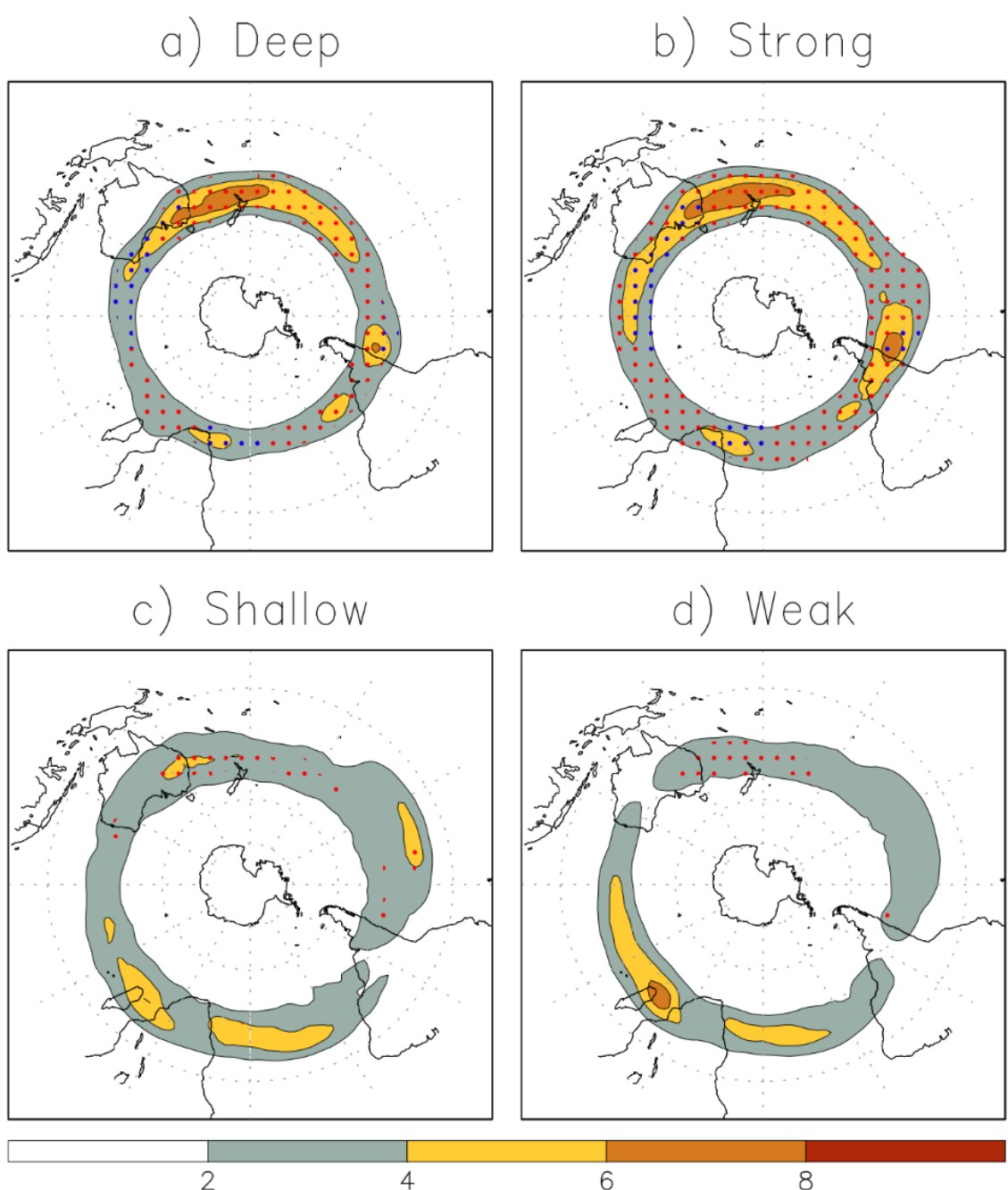

**Figure 1:** Track density (shaded) and mean intensity (dots) for (a) deep, (b) strong, (c) shallow, and (d) weak COLs in the Southern Hemisphere. Track density is measured in number per season per unit area, where the unit area is equivalent to a 5° spherical cap ($\cong 10^6$ km$^2$). Mean intensities less than -8 × 10$^{-5}$ s$^{-1}$ (-12 × 10$^{-5}$ s$^{-1}$) are represented in red (blue) dots.

Figure 2 shows the seasonal variations in the average monthly number, intensity and latitude of COLs for each of the four distinct categories. A clear correspondence in seasonality is evident between deep and strong COLs, as well as between shallow and weak types. Deep and strong COLs exhibit similar intensities (Fig. 2c), though the deepest systems display a more pronounced seasonal cycle in intensity compared to the strongest ones. Deep and strong COLs are more prevalent at 31-33°S (Fig. 2b) from autumn to spring, with two peaks in May and October (Fig. 2a). These peaks appear to be associated with a semiannual oscillation in the polar front jet, albeit with two-month delay from the first peak (Fig. 2d). The half-yearly cycle in the eddy-driven jet is attributed to a response of meridional temperature and pressure gradients between middle and high latitudes which peaks during equinoctial seasons (Van Loon 1967). Our findings align with previous studies and demonstrate a similar cycle to that observed for mid-latitude COLs and Rossby wave breaking events on the 330 K isentropic surface (Ndarana and Waugh 2010, Favre et al. 2012).

In summer, shallow and weak COLs exhibit notably higher frequencies compared to the fewer occurrences observed in winter months. This seasonal pattern appears to be closely linked to the fluctuations in subtropical jet intensity, which experiences a decline (increase) in summer (winter). The increased (reduced) frequency of COLs appears to correspond with the weakened (intensified) subtropical jet strength in summer (winter). This association likely arises because shallow and weak COLs tend to occur more equatorward, roughly coinciding with the subtropical jet position (see Fig. 3). This aligns with the idea that COLs are primarily found over regions of weakened westerly winds (Nieto et al. 1998), and this hypothesis is supported by a robust negative correlation of -0.95 (-0.97) significant at 95% between shallow (weak) COLs and subtropical jet intensity.

Overall, there exists a discernible relationship between upper-tropospheric intensity of COLs and their vertical depth, establishing both classifications as pertinent parameters for assessing the vertical structure of these phenomena. The association of deeper COLs with stronger cyclonic vorticity can be attributed to geostrophic balance and the vertical coupling of atmospheric motions. Enhanced upper-tropospheric circulations induce increased vertical motions and static stability anomalies, resulting in the formation of deeper vertical structures. To maintain consistency and simplicity in our analysis, our subsequent investigations will solely concentrate on the classification based on COL depth.

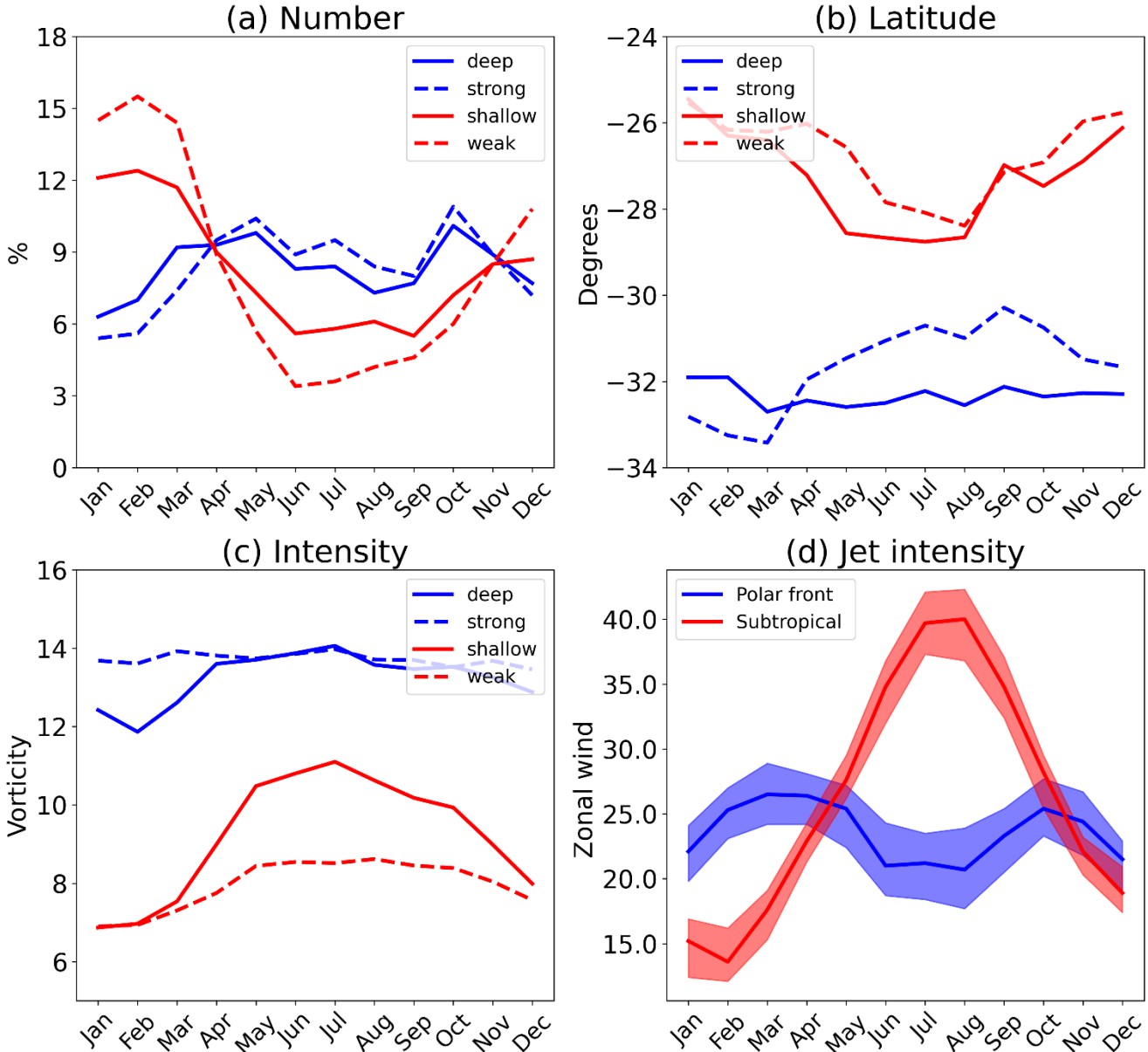

**Figure 2**: Zonal monthly mean characteristics of COLs categorized by deep, strong, shallow and weak for: a) number, b) latitude reached at the time of maximum intensity, and c) maximum intensity at 300-hPa vorticity (scaled by $-1 \times 10^{-5}$ s$^{-1}$). d) Intensity of subtropical and polar front jets as represented by the climatological 300-hPa zonal mean zonal wind at 25°S-35°S and 50°S-65°S, respectively.

**3.3 Seasonal variations in Jet-COL interactions**

The interplay between COLs, jet streams, and Rossby wave breaking has been documented in previous studies (Ndarana and Waugh 2010, Barnes et al. 2021). To unravel this relationship further and elucidate how jets influence COL deepening, we present seasonal mean maps of upper-level zonal winds and track density for deep and shallow COLs, as illustrated in Figure 3. A distinct polar front jet, often referenced as the mid-latitude jet in literature, is apparent from the South Atlantic to the South Pacific oceans throughout the year, exhibiting a poleward spiraling pattern. In the Australia-New Zealand sector, the well-known split jet flow becomes more pronounced during the cool season due to a stronger subtropical jet (Fig. 3c). This pattern results in weak westerlies between the jets, inducing anticyclonic vorticity on the equatorward side of the subtropical jet and cyclonic vorticity poleward (refer to Fig. S2 in Suppl. Material).

Deep COLs occur at more poleward latitudes and typically near the equatorward exit region of the polar front jet. This is particularly noticeable around Australia, albeit with some seasonal and spatial variations. During winter, COLs are less frequent but more intense than other seasons (see Fig. 2), which might be a result of a stronger subtropical jet which generates cyclonic vorticity anomaly over its poleward side. Conversely, the subtropical jet is weak or absent in summer, when a higher occurrence of shallow, weak COLs is observed, particularly over the oceans and closer to the equator compared to deep systems. Interestingly, deep COLs peak in autumn and spring, coinciding with a strengthened polar front jet reaching a comparable magnitude to the subtropical jet. Additionally, during transitional seasons, a secondary maximum in deep COLs is observed near the west coast of South America, particularly in spring when the maximum density occurs with reduced zonal flow at the end of the polar front jet on the poleward side of the subtropical jet.

Our findings support the idea that the subtropical and polar front jets affect COLs differently, aligning with previous observations of their contrasting effects on Rossby wave breaking and COL development (Ndarana and Waugh 2010, Muñoz et al. 2020). While the polar front jet seems to favor the COL deepening, the subtropical jet emerges as a key mechanism for their intensification. However, uncertainty persists regarding the exact relationship between subtropical jet intensity and COL intensity. It is notable that while Rossby wave breaking and COLs manifest over reduced zonal flow which generally occur in regions of weak climatological zonal winds, the mechanisms conducive for COL deepening/intensification are more likely in the proximity of jets. This is supported by earlier observations indicating that a stronger polar front jet leads to a more pronounced dipole pattern with enhanced cyclonic and anticyclonic vorticity anomalies (Bals-Elshol et al. 2001). A more detailed discussion on the influence of jet streams in COLs will be provided in the following sections.

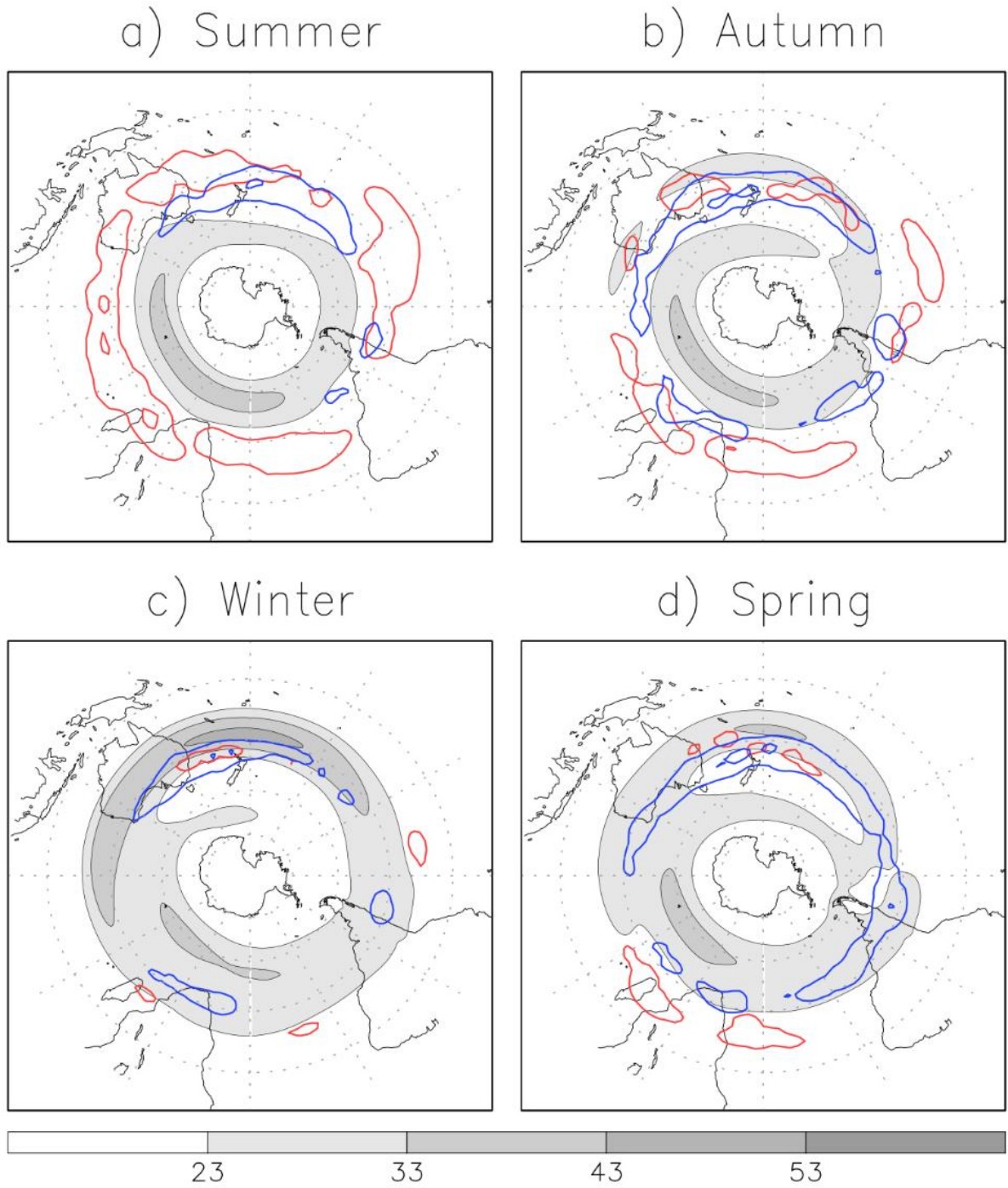

**Figure 3**: Zonal mean wind (shaded) and track density for deep COLs (blue contour) and shallow COLs (red contour) in the Southern Hemisphere for a) Summer (DJF), b) Autumn (MAM), c) Winter (JJA) and d) Spring (SON). Unit is as in Fig. 1 for

track density, and m.s$^{-1}$ for zonal wind. Track densities are plotted for interval contours of 4 units. All fields are represented at the 300-hPa level.

### 3.4 Distinct effects of polar front and subtropical jets on COL depths

To further investigate the influence of jets on COLs, scatter plots are employed to visualize the relationship across different jet intensities, as given in Figure 4. Following the approach detailed in Section 2.4, we found a moderate positive correlation between COL number and polar front jet intensity (r = 0.47 at 99% confidence level), implying that intensifying polar front jet are correlated with increased occurrence of deep COLs. This correlation accounts for approximately 22% of the variability in deep COLs. Shallow COLs also show a positive correlation with the polar front jet intensity, but weaker than that for deep

COLs (r = 0.25, p < 0.01), explaining only 6% of their variability. Considering the subtropical jet (Fig. 4b), no significant relationship with the occurrence of deep COLs is observed. In contrast, a significant negative correlation of 0.75 (99% confidence level) is found for the relationship between the intensified subtropical jet and the reduction in shallow COLs. Approximately 56% of the variability in shallow COLs can be attributed to changes in subtropical jet intensity, aligning with their seasonal variations, as illustrated in Fig. 2.

However, whilst the raw counts show a robust relationship between the intensified subtropical jet and the reduction in shallow COLs (see Fig. 4b), the strength of this relationship weakens when the seasonal cycle is removed from both variables (Fig. 4d). This also observed for the relationship between the polar front jet and deep COLs (Fig. 4c). Hence, whilst there are issues in removing the seasonal cycle as a fixed factor (Pezzulli et al. 2005) the relationship between the strength of the subtropical and polar front jet and COLs remains a hypothesis due to its uncertainty which requires further study.

Our analysis reveals a notable positive correlation between the intensity of both shallow and deep COLs and the subtropical jet (refer to Fig. S3 in supplementary material), but the strength of these relationships weakens when the seasonal cycle is removed. This weakening is likely due to the pronounced seasonal cycle in the subtropical jet compared to the monthly variations from climatological means (as shown in supplementary Fig. S4). Removing the seasonal cycle weakens the signal, posing a limitation in considering the annual cycle as a fixed factor (Pezzulli et al. 2005). This highlights some uncertainty

regarding these relationships, suggesting the need for further work to understand the influence of the jets and their seasonal cycle on the COL intensity. Additionally, no significant relationship is found between the intensity of COLs and the polar front jet (Figure 4c), implying that the polar front jet primarily influences the variability of COLs rather than their intensity.

These differential effects likely arise from the distinct positions and roles of both polar front and subtropical jets within the large-scale atmospheric circulation. The polar front jet, primarily occurring at mid- and high-latitudes, is an eddy-driven jet

associated with enhanced temperature gradients and baroclinicity. In contrast, the subtropical jet, situated around 25°–30°S, arises from the momentum flux of the meridional circulation at the Hadley cell edge which seems to act as a waveguide for Rossby waves breaking associated with 200-hPa COLs typically found at lower latitudes (Muñoz et al. 2020). The results of this study corroborate previous findings regarding the formation of COLs under reduced zonal wind. However, it is important

to emphasize that the presence of a nearby jet is crucial for providing energy necessary for the intensification and/or deepening
of the system. A more detailed discussion on the jet-COL interaction will be provided in the next section.

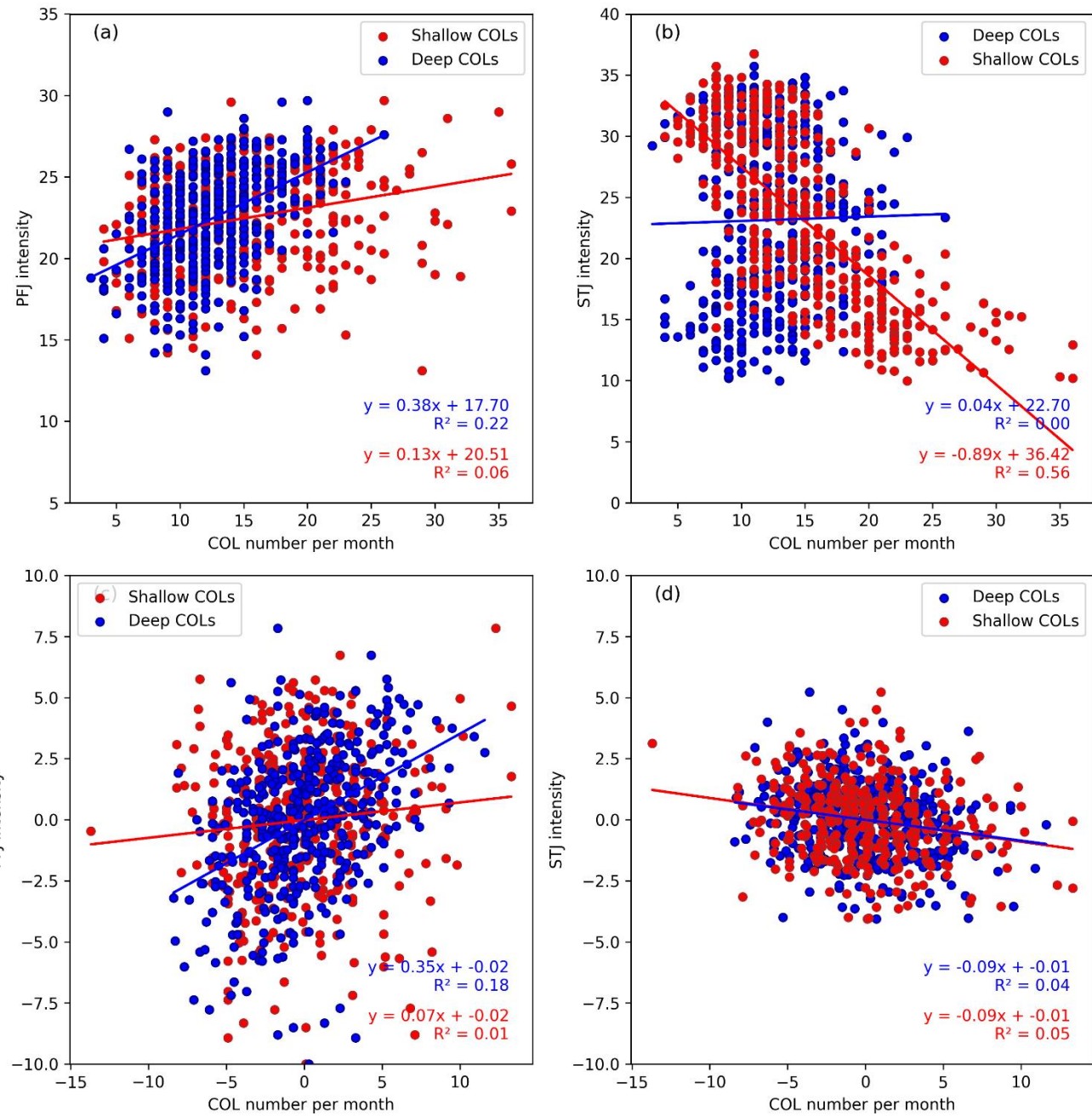

**Figure 4**: Scatter plots indicating the relationships between monthly mean COL number and jet intensity for (a, c) polar front jet and (b, d) subtropical jet using (a, b) raw values and (c, d) anomaly values. Anomalies are calculated by subtracting the

monthly climatological mean from the observed value. Deep and shallow COLs are depicted by blue and red colors, respectively. Unit is in m.s$^{-1}$ for jet intensity.

### 3.5 Shallow COLs vs deep COLs: contrasting from the energetics point of view

When it comes to unraveling the intricate mechanisms behind the deepening of COLs, understanding the role of specific mechanisms in determining their vertical extent can be aided by examining their energetics. Early studies have investigated the dynamical mechanisms of COLs by examining their vertically integrated energy budgets (Gan and Piva 2013, 2016, Ndarana et al. 2020, Pinheiro et al. 2022). In this study, we adopt a similar approach but direct our attention towards the two primary contributing mechanisms of the EKE budget: baroclinic conversion and ageostrophic flux convergence. We compare deep and shallow COLs and investigate the possible implications of energetics for their deepening.

Figures 5 and 6 show the temporal evolution of composite shallow and deep COLs, respectively, for horizontal and vertical fields. The four stages described in the conceptual model of Nieto et al. (2005) and shown in Pinheiro et al. (2021) are seen to be reproduced for shallow and deep COLs, involving the following stages: upper-level trough (-48h), tear-off (-24h), cut-off (0), and decay or dissipation (+24 and +48h). Time is referenced to the time of maximum intensity in the 300 hPa vorticity.

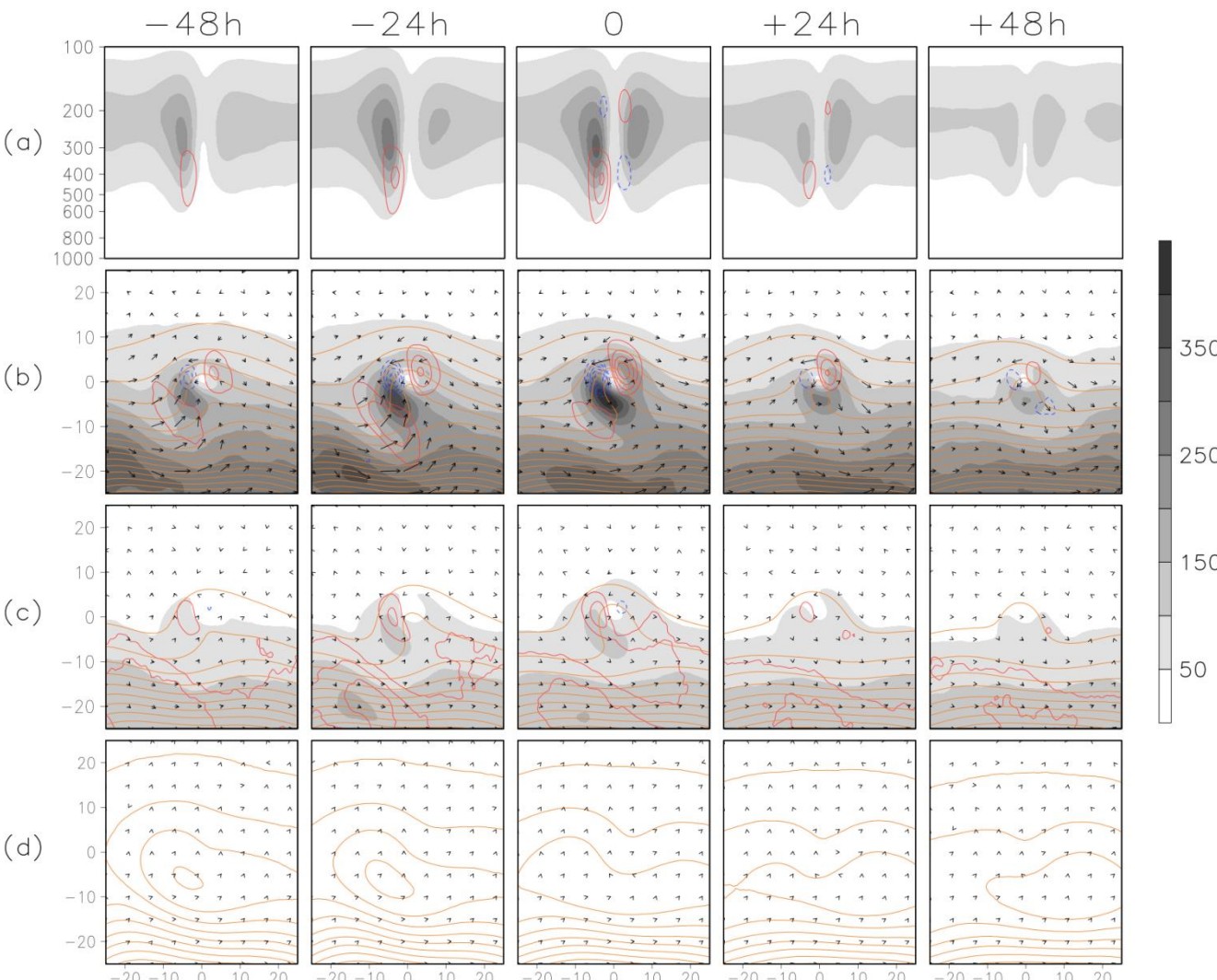

**Figure 5** Temporal evolution of shallow COLs in the Southern Hemisphere relative to the time and space of maximum intensity in $\xi_{300}$. The panels depict: (a) vertical cross-sections of total EKE (shaded) with baroclinic conversion (contour); (b) vertically integrated ageostrophic flux convergence (contour) with EKE (shaded), geopotential height (orange line) and ageostrophic fluxes (vectors) at 300 hPa; (c) vertically integrated baroclinic conversion (red contour) with EKE (shaded), geopotential height (orange line) and ageostrophic fluxes (vectors) at 500 hPa; and (d) EKE, geopotential height (orange line) and ageostrophic fluxes (vectors) at 1000 hPa. Contours represent $0.003 \times 10^{10}$ Joule.s$^{-1}$ for integrated quantities, 50 gpm for geopotential height at 300 and 500 hPa, and 20 gpm for geopotential height at 1000 hPa, while total EKE is indicated by $10^9$ Joule.

In the upper-level trough and tear-off stages of shallow COLs (T = -48h and T = -24h in Fig. 5), a poleward jet located upstream of the ridge axis, i.e., west of the COL, acts as the primary energy source for the COLs. Concurrently, ageostrophic fluxes

transport EKE northeastward from the poleward jet to the rear side of the COLs, forming an energy center downstream of the ridge axis, referred to as the western energy center. As the trough-ridge system deepens in an anticyclonic orientation, the western energy center expands due to the convergence of ageostrophic fluxes and positive baroclinic conversion that arises

downstream of the ridge axis, driven by descending cold air (not shown). The eastward propagation of the poleward jet and its increasing zonal flow give rises to anticyclonic barotropic shear flow and subsequent potential vorticity overturning, as documented in prior studies (Ndarana et al. 2021, Pinheiro et al. 2022). The COL decay occurs when the poleward jet shifts to the east, ceasing to provide energy to the system, as shown in Fig. S5 of Suppl. Material.

Differences in the structure and lifecycle of shallow and deep COLs are observed. While shallow systems exhibit a gradual

weakening and anticyclonic circulation at the surface, deep COLs display a multi-level interconnected vortex structure, as illustrated in Figure 6. During their development phase, deep COL systems exhibit stronger ageostrophic fluxes along the ridge axis, and larger baroclinic conversion on the upstream poleward jet compared to shallow systems. The positioning of deep COLs near the equatorward exit region of a strong polar front jet (as depicted in Fig. 3) intensifies the convergence of ageostrophic fluxes, thereby increasing baroclinic conversion. Consequently, this amplification leads to heightened vertical

motions and deeper circulation. This distinct feature is prominently observed within deep COLs, distinguishing them from their shallow counterparts.

Another distinct aspect of deep COLs arises during the cut-off stage (T = 0 onward), when enhanced vertical motions drive increased baroclinic conversion towards the east and near the surface. The interaction between upper- and lower-tropospheric eddies, as in mid-latitude baroclinic waves (Hoskins and Karoly 1981, Trenberth 1991, Nakamura 1992), suggests an eddy

feedback mechanism between the eddy-driven jet and lower-level thermal forcing (Kushner et al. 2001, Deser et al. 2004, Lu et al. 2014). This feedback likely arises due to thermal wind adjustment (Ring and Plumb 2007, Nie et al. 2016). Additionally, downward eddy momentum activity fluxes also likely contribute to vertical energy propagation, as observed in earlier studies (Trenberth 1991, Rivière et al. 2015).

During the decay stage of deep COLs, ageostrophic fluxes and eddy feedback mechanisms facilitate the downstream export of

EKE, maintaining and shifting the jet eastward (refer to Fig. 6 at T = +48). The stronger baroclinic processes and ageostrophic fluxes observed in deep systems explain their longer lifetimes (4.7 days) compared to shallow COLs (3.3 days), due to a greater interaction between mechanisms operating at various levels within these systems.

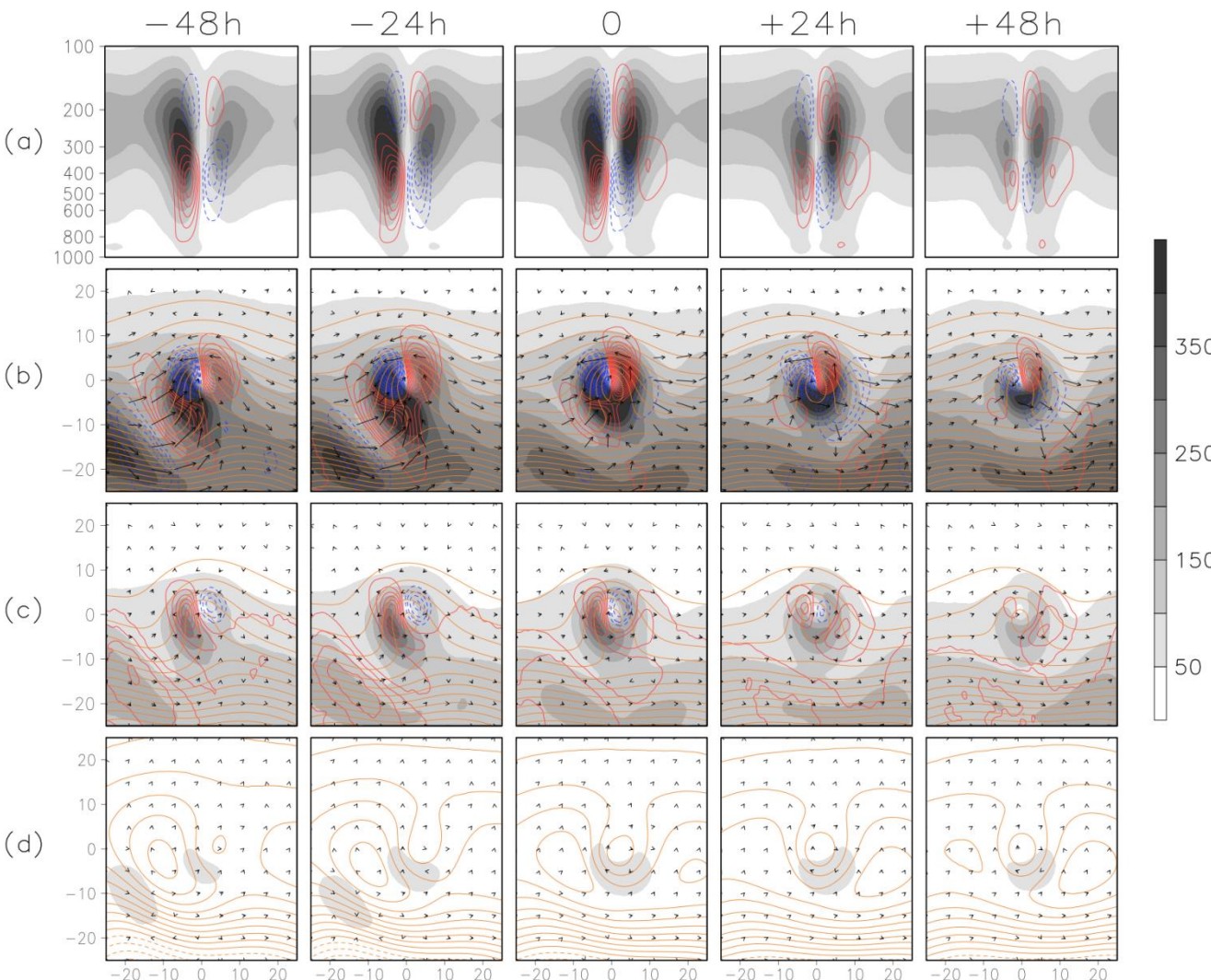

**Figure 6:** Same as Figure 5 but for deep COLs in the Southern Hemisphere. Dashed lines at the bottom indicate negative geopotential height at 1000 hPa.

A key feature of our approach lies in its consideration of pre-existing low-level cyclones linked to COLs. This is achieved by using a relatively short temporal threshold for matching, which significantly expands our ability to detect a wide range of multi-level stacked lifecycles. While the results exhibit some sensitivity to the chosen method, they remain remarkably consistent with previous observations of rapid vertical evolution of potential vorticity cut-offs, which typically reach their maximum extent roughly one day after genesis (Portmann et al. 2021). Furthermore, the observed characteristics of deep COLs closely align with patterns identified in Catto (2018) for Cluster 3 and Sinclair and Revell (2000) for Class T in the Australia and New Zealand region, with a cyclone originating directly beneath a deep upper-level trough or a cut-off potential vorticity streamer.

**3.5 Impacts of diabatic processes on residual energy and its influence in the deepening of COLs**

Numerous studies have demonstrated the influential role of diabatic processes, such as radiation, latent heating and planetary boundary layer processes, in the development of synoptic-scale systems (Davis and Emanuel 1991, Stoelinga, 1996). As pointed out above and discussed in some detail by Pinheiro et al. (2022), inaccuracies arising from the misrepresentation of diabatic heating in reanalyses can introduce uncertainties into the energetic framework. Therefore, investigating the influences of diabatic processes on residual energy can enhance our understanding of their impact on the deepening mechanisms of COLs.

Figure 7 shows the spatial and temporal distribution of residual energy in composites of shallow and deep COLs. The patterns of residual energy exhibit similarities in shallow and deep COLs, with significantly higher magnitudes observed in deep systems. Negative residual energy dominates throughout the lifecycle of both shallow and deep COLs, as indicated by the red values in Figure 7, representing the residual integrated volume within a 15-degree radial distance centered on the vortex center.

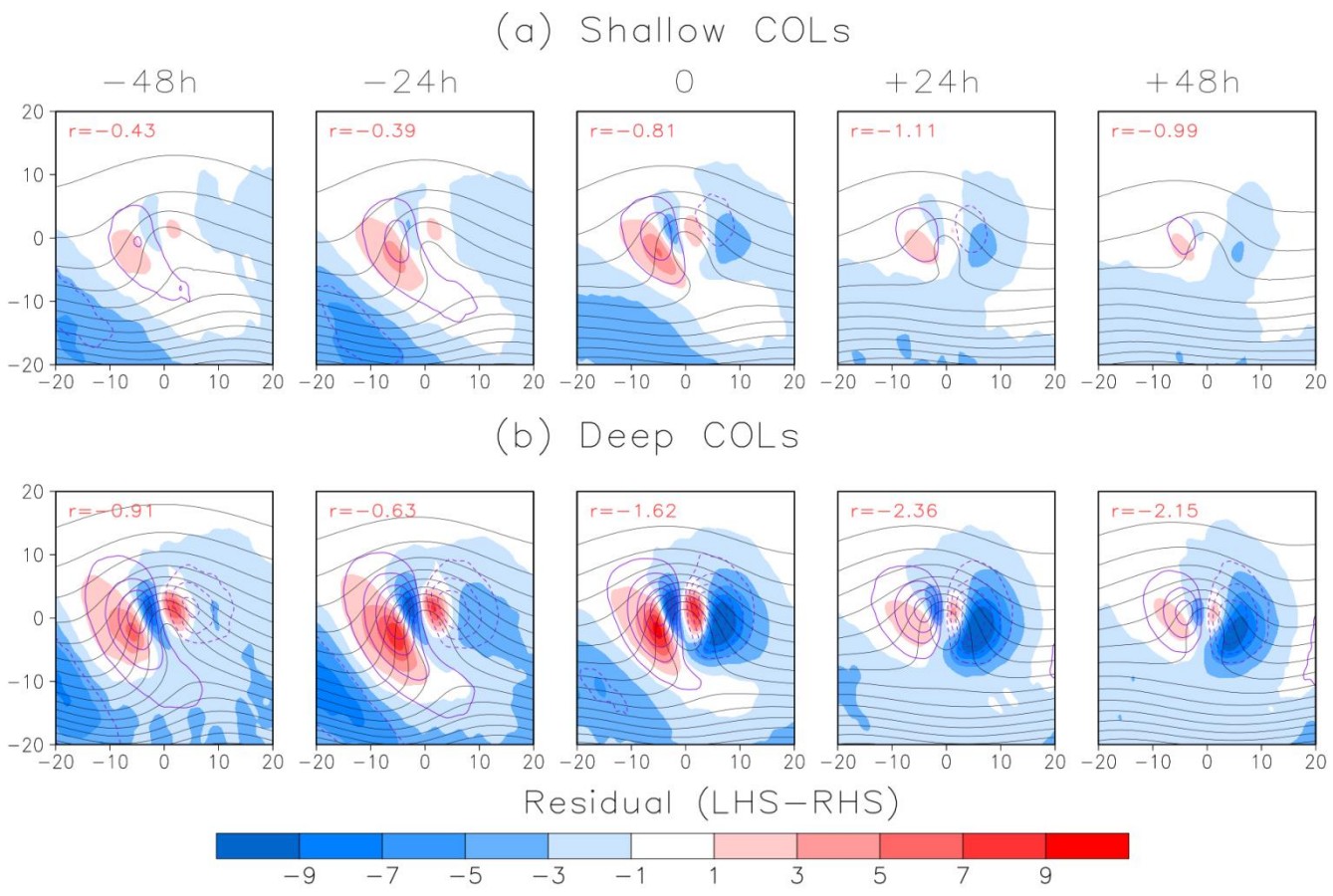

**Figure 7:** Temporal evolution of the residual energy in Joule (shaded) scale by $10^9$, geopotential height (black contour) at 300 hPa for contour intervals 50 gpm, and vertical velocity (purple contour) for contour intervals 0.05 Pa.s$^{-1}$, where solid (dashed)

contours indicate positive (negative) values. Red values represent the residual energy vertically averaged within a 15-degree spherical cap region centered on the COL location.

While negative residual energy prevails, it is worth noting that positive values emerge west of the COLs during the upper-level trough and tear-off stages. Conversely, negative residual energy is observed to the east of the COLs during the cut-off and decay stages. This contrasting pattern is particularly pronounced in deep COLs, consistent with previous findings on strong COLs (see Figure 4 of Pinheiro et al. 2022).

The residual energy suggests the existence of development mechanisms that are either not considered in our approach or inadequately represented in the reanalysis data. A common issue lies in accurately representing diabatic processes, such as radiative cooling and latent heating, which pose challenges for reanalysis. A significant negative residual, particularly pronounced in deep COLs, is observed east of COLs where enhanced ascent and convection can introduce additional energy sources and unresolved phenomena contributing to the residual. In contrast, positive residual prevails along the western COL edge, possibly influenced by sinking air that promotes radiative cooling.

Uncertainties in measurement techniques, including inhomogeneous observations, analysis increment, and model parameterizations, are especially important in regions of intense convection. Although diabatic heating does not directly influence the EKE budget as it is not included in the energetic framework, errors in the reanalysis due to the misrepresentation of these processes (see https://confluence.ecmwf.int/display/FUG/Section+4.2+Analysis+Increments) can lead to inconsistencies in the energetic analysis, as discussed by Pinheiro et al. (2022). Our results suggest that COLs that are more strongly dependent on diabatic processes are less well represented by reanalysis data. Additionally, the contribution of friction remains an unknown factor, challenging to quantify directly. It is possible that stronger systems have a heightened residual contribution from frictional effects, although this remains an area of ongoing investigation.

## 4 Discussion and conclusions

This study has introduced a track matching algorithm applied to ERA-Interim reanalysis for accurate estimation of COL depth. Our findings highlight that the accuracy of COL depth estimation is primarily influenced by the temporal overlap ($\chi$) between tracks at different pressure levels. We found that employing $\chi = 1\%$ with a mean separation distance of 5° geodesic provides a feasible approach for capturing COLs during specific lifecycle stages. Notably, our method reveals a lower proportion of deep COLs compared to previous studies (Porcù et al. 2007, Barnes et al. 2021). This discrepancy arises from differences in methods, where our vorticity-based approach detected a larger number of systems, leading to a relatively smaller frequency of deep COLs. However, vorticity-based identification offers advantages over geopotential data as it avoids subjective threshold adjustments across pressure levels and is more sensitive to weaker COLs, providing a more consistent method for analyzing their vertical structure.

Further, we investigated the contrasting characteristics of COLs based on their vertical depth and intensity, categorizing them into four main types: shallow, deep, weak, and strong. Deep and strong COLs exhibit similar distribution patterns,

predominantly concentrated in Australia and the southwestern Pacific. They are more intense and frequently found poleward of the subtropical jet, displaying a semi-annual oscillation peaking in autumn and spring. In contrast, shallow and weak COLs are less intense, situated more equatorward and with increased frequencies in summer. However, despite these similarities, approximately 20-30% of deep and shallow COLs align with the 50th percentile of weakest and strongest systems, respectively. The variability within these classifications underscores the complexity inherent in COL dynamics, highlighting the need for future research to explore the interdependencies among these categories.

A particular novel aspect of our study is the differential impact of both subtropical and polar front jets on COLs. While COLs typically originate in regions characterized by weak zonal flow, the proximity of a jet streak is crucial for the system intensification and/or deepening. Our findings reveal that the COL deepening is more probable in the presence of a robust upstream polar front jet, due to enhanced convergence of ageostrophic fluxes and baroclinic conversion, resulting in intensified vertical motions and a deeper cyclonic circulation. However, a stronger polar front jet does not necessarily correlate with more intense COLs.

Conversely, a positive correlation exists between COL intensity (both shallow and deep) and the subtropical jet, although this weakens when considering the seasonal cycle. This suggests limitations in using the annual cycle as a fixed factor. The COL intensification might be induced by the wind shear along the subtropical jet edge, which is supported by the observed small-scale jet stream equatorward of COLs (Ndarana et al. 2021). Nevertheless, uncertainties remain regarding the causal nature of this relationship. Further investigations employing causal inference methods (e.g., Samarasinghe et al. 2019, Docquier et al. 2024) could be employed to establish the cause-and-effect connections between COLs and jet streams.

Our findings suggest that the deepening process is also influenced by diabatic processes which enhance vertical motions and lead to increased baroclinic conversion around the COLs. These processes contribute to longer lifetimes and greater interaction between mechanisms operating at different levels within deep COL systems. However, the misrepresentation of diabatic processes in reanalysis is a particular issue that affect the residual energy, leading to uncertainties. Negative residual energy dominated throughout the COL lifecycle, notably higher in deep systems. The presence of negative (positive) residual energy east (west) of the COLs points to the existence of dissipation (intensification) mechanisms not adequately represented in the reanalysis data, possibly related to latent heating (radiative cooling), as observed in Cavallo and Hakim (2010). Accurately representing diabatic processes in reanalyses is essential to maintain consistency in the energetic analysis and understand their role in deepening COLs. Therefore, it is crucial to re-evaluate COLs and the EKE budget whenever new reanalysis products such as ERA5 become available in future work.

In summary, our research emphasizes the importance in classifying COL for understanding their key features and reveals their intricate relationships with jet streams. However, further development of the energetic framework, incorporating vertical ageostrophic fluxes as in Rivière et al. (2015), remains essential for comprehensively elucidating the eddy feedback mechanisms.

## Acknowledgements

We would like to thank CNPq (Conselho Nacional de Desenvolvimento Científico e Tecnológico, Grant: 151225/2023-0) and FAPESP (Fundação de Amparo à Pesquisa do Estado de São Paulo, Grant: 2023/10882-2) for their support and funding throughout this research. The authors have employed artificial intelligence (AI) to enhance the quality of the paper's writing.

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

## Appendix

See Table A.1.

**Table A.1** Sensitivity of the number of $\xi_{300}$ COLs to time overlap for each pressure level (hPa), expressed as a percentage of the total number. The overlap thresholds are 1%, 5%, 10%, 25%, 50%, 75%, and 100%.

|      | 1%    | 5%    | 10%   | 25%   | 50%   | 75%   | 100%  |
|------|-------|-------|-------|-------|-------|-------|-------|
| 300  | 100.0 | 100.0 | 100.0 | 100.0 | 100.0 | 100.0 | 100.0 |
| 400  | 80.9  | 80.9  | 80.9  | 80.8  | 79.2  | 66.0  | 11.2  |
| 500  | 65.5  | 65.5  | 65.5  | 65.1  | 61.9  | 42.9  | 2.2   |
| 600  | 51.1  | 51.0  | 50.9  | 50.3  | 45.7  | 25.8  | 0.5   |
| 700  | 37.7  | 37.7  | 37.5  | 36.9  | 31.6  | 14.9  | 0.2   |
| 800  | 28.8  | 28.8  | 28.6  | 28.0  | 23.0  | 9.7   | 0.1   |
| 900  | 23.1  | 23.0  | 22.9  | 22.3  | 18.1  | 6.9   | 0.0   |
| 1000 | 20.3  | 20.2  | 20.1  | 19.5  | 15.4  | 5.6   | 0.0   |

## Code/data availability

The code used in this study is available upon request from the corresponding author for researchers interested in reproducing or extending the findings presented in the paper. The ERA-Interim reanalysis data was sourced from the ECMWF server. Please contact corresponding author's email for access to the code.

## Author Contributions

Henri Pinheiro: Conceptualization, investigation, data curation, methodology, writing – original draft, visualization.

Kevin Hodges: Software development, writing - review and editing.

Manoel Gan: Writing - review and editing.

## Competing interests

The authors declare no competing interests in relation to the research, authorship, or publication of this paper.