# Peer review of "Deepening mechanisms of cut-off lows in the Southern Hemisphere and the role of jet streams: insights from eddy kinetic energy analysis"

_EGUsphere, 2023_

## Referee Comment (RC1)

Review of "Exploring the vertical extent and deepening mechanisms of cut-off lows in the Southern Hemisphere: insights from eddy kinetic energy analysis" by Pinheiro et. al.
* * *
Suggested outcome: Major Revisions

Scientific significance: Good
Scientific quality: Good
Presentation quality: Good
* * *
This work creates a climatology of cut-off low depth over the Southern Hemisphere and studies the vertical extent and mechanisms that lead to deep cut-offs from an energetics perspective.
* * *
**General comments:**
The authors present a different methodology for tracking cut-off depth and vertical extent using an established cyclone tracking algorithm. Although the methodology and climatology are relatively rigorous, there seems to be little discussion of the mechanisms and processes that lead to cut-off vertical extent that the authors pose. I do concede that the authors refer to arguments made in previous work, but these need to be fleshed out more and discussed more fully here for the reader to understand their arguments. Importantly, I feel there are still gaps in the evolution of the coupled upper-lower tropospheric processes.

**Major comments:**
1. Upper-level processes in relation to the lower-level processes
   The methodology and the results of the cut-offs in relation of the lower-level processes are obviously critical to the results of this work. There however appear to be some gaps in the authors arguments as to how well the methodology captures this link and/or separation. The authors should consider expanding on this process to enhance the value of this work.

   The authors use a top-down approach when searching for vertical extent of cut-offs. This is a sensible choice of course. However, the authors also admit that this approach may not capture all coupling types. Do the authors see any evidence of other coupling types in the data they have collected? For example, Figure 5 shows a closed surface circulation at the "upper-level trough" phase (T-48). Does this show evidence that the surface is developing and closing prior to the cut-off and thus is developing from the surface, upwards towards the upper troposphere? Or that the cut-off enters a region of a pre-existing surface low? Is discussion of what occurs prior to T-48 required to explain the potential differences in deep and shallow cut-offs, since the cyclonic circulation seems relatively mature (although not cut-off) by T-48? Additionally, the shallow cut-off composites in Figure 4, show that some degree of surfaceward extension is occurring since there a cyclonic zone, albeit weak, at the surface. Is all we are seeing simply an intense (for deep) versus weak (for shallow) cyclonic circulation in the upper levels with "action at distance"? If so, are the dynamical processes really that different?

   Further to this, the authors suggest that the decrease in tracks when expanding the requirement for temporal coherence suggests that the coupling is most frequently in the mature phase. Could an argument not be made that this decrease could be the result of their independence from one another. Ie. could the larger number of extended COLs that occur with a small temporal coherence could result from many COLs simply moving over a low-level baroclinic zone or pre-existing low-level cyclone?

2. Depth of dynamical reasonings
   Figures 4 and 5 are great, but the discussion of them and the processes at play are never really

fully discussed. One should really go into detail in the framework chosen as to how these processes play out.

Often dynamical reasons are brief and simply reference the authors previous work. This is fine of course, however, I found it difficult to follow some of these arguments and reasonings without jumping between several different papers. The manuscript would be fleshed out significantly by extending and fleshing out some of these arguments somewhat to provide a fuller picture to the reader.

**Specific comments:**

- L33: "high potential vorticity anomalies" – ambiguous in the southern hemisphere as there we deal with large negative values of PV. Suggest the use of "large magnitude" or "cyclonic".
- L57: "ageostrophic fluxes" is used throughout the manuscript. Is "ageostrophic geopotential fluxes" a more accurate description of this term?
- L57-L62: Use of multiple adverbs started sentences in a row (ie. "Furthermore,…" and "Additionally,…"). Suggest to rewrite so that this paragraph flows more easily.
- L76: Is there a reason the authors are not using the latest reanalysis (ERA5)?
- L82: "similarly as done before" -> "as done in previous work"?
- Methods: The authors explain throughout the manuscript the advantage of vorticity tracking to include small-scale cyclonic circulations. Is there a sensitivity of the choice of 5-degrees when looking at whether that circulation is closed? I.e. is it possible if the vorticity minimum is small-scall for the u and v components to be unrelated to the cyclonic circulation identified?
- L136: "It could also" -> "Errors could also…"?
- Figure 1: Panel b) is labelled as panel a) in the figure title
- Figure 1: The most intense density of COLs is located on the Mozambiquan channel. As the authors use a "cyclonic circulation only" type tracking without taking into account core temperatures, are the authors picking up transitioning Tropical Cyclones in this region?
- Figure 2: The presentation of these results as well as some of the wording in the explaining paragraphs (ie. L209-211) could be improved to make the point of extension to low-levels without extension to the surface clearer. The "sharp decreases" in regions A and C (L211) are difficult to see.
- Figure 3: It may be useful to plot some proxy for the jets on this figure as this is a large part of the authors argument for why deep COLs preferentially occur in specific regions. Does the seasonality of these COL depths coincide with when the split jet occurs (during the cool season)? This discussion should also be expanded.
- L243: "Figure 1c" -> "Figure 3c"?
- L242-243: "southeastern Pacific, where deep COLs observed at more northern latitudes" – there doesn't seem to be that much change in latitude from Figure 3. Consider some latitude statistics to prove this point.
- Figure 4 and Figure 5 - do both of the timesteps provided represent the relevant phases that the author suggests in L254-255? For example shallow COLs at T0 seem to be similar (at least in the upper-levels) to deep COLs at T-48? Do shallow COLs actually ever reach maturity?
- Figure 6: Deep cut-offs appear to be embedded somewhat in really strong westerlies? Is this true? And does this have an impact on the associated baroclinicity? This point is very briefly mentioned (L276), but could be expanded on.

---

## Referee Comment (RC2)

Comments on "Exploring the vertical extent and deepening mechanisms of cut-off lows in the Southern Hemisphere: Insights from eddy kinetic energy analysis

By Pinheiro HR, Hodges KI and Gan MA.

The study considered COLs in the SH and then categorized them according to how deep they. These categories were shallow, medium and deep COLs and the latter were shown to extend to the surface. The study further undertakes an energetics analysis to gain insights into the mechanism that could be responsible for the extension of COLs to the surface. This is a well written and succinct paper. It has the potential of making a contribution to the current work that is ongoing on COLs in the SH. I recommend that it be considered for publication, provided the comments below are adequately addressed.

Comments:

2. Eq 1. This study aims to consider the three dimensional structure of COLs, so why are the authors then taking the volume integral? This will average our processes that I believe are important to consider if the vertical structure is to be assessed. So, I challenge the authors to consider the EKE that is not integrated to reveal vertical processes (as will be mentioned again later in this review). So I am suggesting that consider the diagnostics used here carefully.

3. Top panels of Fig 4 and 5. How were the vertical profiles produced? Did you average over a range of latitudes.
   Please include the jet in this plot, they are, in my opinion, an important piece of the puzzle.

4. Paragraph 270: I agree with the notion that the eddy driven jet is a primary source of EKE and please use the ridge axis as a reference to differentiate between the energy centre that develops from baroclinic conversion and the one that arises as a result of ageostrophic flux convergence. Recent studies in the SH (even though they have focused on regional issues) have shown that there is very weak or no baroclinic conversion downstream of this ridge axis. The authors should explain the cause for the vertical circulation that leads to baroclinic conversion. This is particularly important in this paper because it looks like the authors argue that the midlatitude jet streak is responsible for differences between shallow and deep COLs, and as a reviewer, I totally agree with that.

5. Paragraph 275: I think that question of why the jet is stronger for deeper COLs must be addressed.

   Still on this issue: The authors mention and show that the ageostrophic fluxes are stronger for deeper COLs. This makes perfect sense but why? Would the authors agree with the hypothesis that a stronger jet causes increased anticyclonic barotropic shear, which in turn causes higher strain rates and therefore the higher likelihood for RWB (Nakamura and Plumb 1994)? This means that the flow across the ridge axis becomes more super geostrophic, implying a stronger ageostrophic flow across the ridge. But the breaking increases the intensity of the geopotential height anomalies. Thus, combining these two issues leads to stronger ageostrophic fluxes and stronger EKE in the COLs.

I also curious about the ageostrophic fluxes associated with the sub-geostrophic flow that the categorization suggested here appears to be revealing. What does that mean for the energy centre at the inflection point immediately downstream of the ridge axis? No mention is made of this and it looks like it is playing some role.

Why would a stronger jet streak lead to stronger baroclinic conversion: The authors could invoke jet streak dynamical theory to explain this. For instance the cross front quasi-geostrophic theory of Keyser and Shapiro (1986) could used to explain this. Also the location of the baroclinic conversion relative to the jet should be considered to link the strength of jet and the strength of the conversion. The authors should consider the possibility that the midlatitude jet might be curved, which has implications for vertical motion.

6. In point 1 above an issue about the diagnostics used was raised. EKE is not visible at the surface for shallow COLs, that is reasonable. They are however very clearly in existence for deep COLs, even though they are weaker than aloft. So, what causes them? Where does this energy come from? That is why taking the volume integral of the energetics terms is not the appropriate thing to do here. I invite the authors to have a look at Fig 7 in Reviere et al (2015). There is a clear downward flux of EKE and yet the authors say that this flux is small; yet this seems important in explaining where the surface EKE might be coming from.

In conclusion, I think that the use of energetics has a lot of potential in explaining what causes some COLs to extend to the surface and others not. However I am of the opinion that the energetics framework used here is not adequate and I invite the authors to employ the energy equation derived by Reviere et al (2015), without the volume integration to reveal the vertical structure, in particular attempt to explain where the EKE at the surface is coming from. That is lacking in the manuscript.

My suspicion is that deeper COLs last longer and as the authors have raised dissipation issues here, barotropic conversion must be considered.

Therefore, the first two research questions have been addressed, but the third one has not been adequately addressed and I invite the authors to consider the points made above.

---

## Author Comment (AC1)

**Authors' response to reviewers' comments**

Manuscript Number: egusphere-2023-1996

Exploring the vertical extent and deepening mechanisms of cut-off lows in the Southern Hemisphere: insights from eddy kinetic energy analysis

New suggested title:

**"Deepening mechanisms of cut-off lows in the Southern Hemisphere and the role of jet streams: insights from eddy kinetic energy analysis"**

The authors gratefully acknowledge the insightful feedback from the anonymous reviewers and the editor, which significantly strengthened the revised manuscript. We have carefully considered each comment and implemented substantial changes to address them, particularly regarding the influence of jet streams on COLs. We believe the revised manuscript is now significantly stronger and more impactful.

We have made adjustments by removing certain figures (or relocating them to the Supplementary Material) and shortening the text to prevent excessive lengthening of the paper. Furthermore, we have replaced the term "mid-latitude jet" with "poleward jet" during the discussion, as it is a more precise description.

In this document, the responses to reviewers' comments are highlighted in red font.

**Reviewer 1**

**General comment**

The authors present a different methodology for tracking cut-off depth and vertical extent using an established cyclone tracking algorithm. Although the methodology and climatology are relatively rigorous, there seems to be little discussion of the mechanisms and processes that lead to cut-off vertical extent that the authors pose. I do concede that the authors refer to arguments made in previous work, but these need to be fleshed out more and discussed more fully here for the reader to understand their arguments. Importantly, I feel there are still gaps in the evolu3on of the coupled upper-lower tropospheric processes.

**Major comments**

**1. Upper-level processes in relation to the lower-level processes**

The methodology and the results of the cut-offs in relation of the lower-level processes are obviously critical to the results of this work. There however appear to be some gaps in the authors arguments as to how well the methodology captures this link and/or separation. The authors should consider expanding on this process to enhance the value of this work.

The authors use a top-down approach when searching for vertical extent of cut-offs. This is a sensible choice of course. However, the authors also admit that this approach may not capture all coupling types. Do the authors see any evidence of other coupling types in the data they have collected? For example, Figure 5 shows a closed surface circulation at the "upper-level trough" phase (T-48). Does this show evidence that the surface is developing and closing prior to the cutoff and thus is developing from the surface, upwards towards the upper troposphere? Or that the cut-off enters a region of a pre-existing surface low? Is discussion of what occurs prior to T-48 required to explain the potential differences in deep and shallow cut-offs, since the cyclonic circulation seems relatively mature (although not cut-off) by T-48? Additionally, the shallow cutoff composites in Figure 4, show that some degree of surfaceward extension is occurring since there a cyclonic zone, albeit weak, at the surface. Is all we are seeing simply an intense (for deep) versus weak (for shallow) cyclonic circulation in the upper levels with "action at distance"? If so, are the dynamical processes really that different?

Thank you for raising these critical points. We recognize that the original version lacked explanation, in this revision we have attempted to improve this.

Employing a single direction search scheme may not capture all types of coupling, but this does not necessarily invalidate our findings. COLs interact with lower-level cyclonic features in a variety of complex ways, resulting in different coupling patterns.

More advanced methodologies, such as that used by Lakkis et al. (2019), have the potential to offer a more comprehensive understanding of all staked cyclones in the atmosphere, but this is not particularly our goal. We focus specifically on the upper-level forcing driving surface cyclone development (interconnected vortex structure) and the mechanisms driving this interaction. While we may not have sufficiently clarified this in the first version, we have emphasized this focus in our revised manuscript (for detail, please see Section 2.2).

Still related to the methodology, using a vorticity-based feature tracking method with less restrictive temporal overlaps allows us to capture as much of the stacked lifecycles as possible, irrespective of the bottom-to-top or top-to-bottom orientation. This aligns with the findings of Lakkis et al. (2019), who obtained similar though not exactly identical results using both stacking approaches for multi-levelled events. This is also discussed in Section 2.2.

Another aspect of our approach is the possibility of including preexisting cyclones due to the relatively small temporal threshold used for matching. This allows the detection of a broader range of multi-level stacked lifecycles. Techniques based on clustering could potentially provide insights on the spatial patterns of interaction between upper- and lower-level cyclones, as well as the different upper-tropospheric flow patterns associated with upper-level jets, such as equatorward/poleward entrance/exit regions. Some studies (e.g. Sinclair and Revell 2000; Studholme et al. 2015; Catto 2018) have explored these patterns, and this could be investigated in further work. In the revised version, we discuss and compare our current findings with those from earlier studies.

We considered examining events prior to T = -48 hours to identify the atmospheric precursor mechanisms. However, extending the compositing window beyond this timeframe introduces noise that arises from differences in COL lifetimes. Alternatively, considering tracks with lifetimes exceeding four or five days would significantly limit the sample size available for compositing.

We acknowledge that our discussion of the similarities and differences between shallow and deep COLs could be more detailed. However, our revised manuscript does explore their developmental mechanisms in more detail, including a deeper discussion on the role of jet streams for COL deepening.

Further to this, the authors suggest that the decrease in tracks when expanding the requirement for temporal coherence suggests that the coupling is most frequently in the mature phase. Could an argument not be made that this decrease could be the result of their independence from one another. Ie. could the larger number of extended COLs that occur with a small temporal coherence could result from many COLs simply moving over a low-level baroclinic zone or preexisting low-level cyclone?

You raise a valid point. While the decrease in track counts does not definitively establish that coupling primarily occurs during the mature phase, it suggests that COLs likely associate with lower-level features for a limited time. This is consistent with the increased interactions observed when relaxing the threshold. We have reorganized the text and removed the contested sentence, acknowledging the lack of conclusive evidence.

Regarding multiple COLs acting with a single cyclone, we cannot definitively dismiss the possibility, but based on current knowledge, it seems less probable. This interpretation aligns with our understanding of these atmospheric phenomena.

**2. Depth of dynamical reasonings**

Figures 4 and 5 are great, but the discussion of them and the processes at play are never really fully discussed. One should really go into detail in the framework chosen as to how these processes play out.

Often dynamical reasons are brief and simply reference the authors previous work. This is fine of course, however, I found it difficult to follow some of these arguments and reasonings without jumping between several different papers. The manuscript would be fleshed out significantly by extending and fleshing out some of these arguments somewhat to provide a fuller picture to the reader.

Thank you for the feedback. We agree that the discussion of these figures and the processes at play could be more detailed. We have expanded the discussion in the revised manuscript to provide a more detailed analysis.

Also, we understand that it may be difficult for readers to follow our arguments without having read our previous work. We provided some additional information as supplementary material to help readers understand our findings without consulting multiple references.

**Specific comments**

• L33: "high potential vorticity anomalies" – ambiguous in the southern hemisphere as there we deal with large negative values of PV. Suggest the use of "large magnitude" or "cyclonic".

Agree, this has been changed.

• L57: "ageostrophic fluxes" is used throughout the manuscript. Is "ageostrophic geopotential fluxes" a more accurate description of this term?

To avoid confusion, we included the term originally proposed by Orlanski in the first citation, although both terms have the same physical interpretation and have been used interchangeably.

• L57-L62: Use of multiple adverbs started sentences in a row (ie. "Furthermore,…" and "Additionally,…"). Suggest to rewrite so that this paragraph flows more easily.

This sentence was rewritten for clarity.

• L76: Is there a reason the authors are not using the latest reanalysis (ERA5)?

This work was originally started sometime ago before ERA5 was available and used ERA-Interim data to identify COLs. To be consistent with our previous work which used the same COL dataset we have continued with ERA-Interim based analysis. Whilst it would certainly be of interest to repeat this study and our previous studies using ERA5 instead doing this would take some time to complete all the calculations. We hope to continue the work using ERA5 reanalysis in the future.

• L82: "similarly as done before" -> "as done in previous work"?

It has been changed as suggested.

• Methods: The authors explain throughout the manuscript the advantage of vorticity tracking to include small-scale cyclonic circulations. Is there a sensitivity of the choice of 5-degrees when looking at whether that circulation is closed? I.e. is it possible if the vorticity minimum is small scall for the u and v components to be unrelated to the cyclonic circulation identified?

Good point. The thresholds used to identify cyclonic systems are determined based on the type of cyclonic systems typically observed in the region, supported by a limited sensitivity analysis. In general, decreasing the distance from the vorticity center reduces the lifetime and/or the number of identified tracks, meaning that some COLs will be missed.

Objective methods work well for COLs with more symmetric circulation, but some issues arise with tilted troughs, particularly in the early stages. The problem described above can be minimized by avoiding relatively small distances for the wind direction search. However, the authors recognize that this is a difficult task, as the method to identify COLs is somewhat arbitrary.

• L136: "It could also" -> "Errors could also…"?

It has been changed.

• Figure 1: Panel b) is labelled as panel a) in the figure title

We opted to relocate this plot to the supplementary material to prevent elongation of the paper.

• Figure 1: The most intense density of COLs is located on the Mozambiquan channel. As the authors use a "cyclonic circulation only" type tracking without taking into account core temperatures, are the authors picking up transitioning Tropical Cyclones in this region?

We agree it is possible that some of the identified COLs in the Mozambique channel are transitioning tropical cyclones, however, we believe that the majority of these COLs are distinct phenomena.

Note that the identification based on circulation is a post tracking step, the tracking of all systems is done first and then the identification. In a sensitivity analysis, a cold-core condition has also been imposed to the identified tracks as a post-tracking step. This is done by searching a temperature minimum over a spherical cap region within a 5.0° geodesic radius from the vorticity center, as detailed in Pinheiro et al. (2017). Figure 1 shows that the track density is reduced by adding a cold-core condition, but the spatial pattern looks quite similar to the standard method with only winds. This is expected as most tracked systems are essentially cold air cut-offs.

Pinheiro, H. R., Hodges, K. I., Gan, M. A., & Ferreira, N. J. (2017). A new perspective of the climatological features of upper-level cut-off lows in the Southern Hemisphere. Climate Dynamics, 48, 541-559 (https://link.springer.com/article/10.1007/s00382-016-3093-8)

[Figure]

**Figure 1**: Annual track density of SH COLs based on the 300-hPa relative vorticity using (a) 300-hPa horizontal wind components only and (b) 300-hPa horizontal wind components with 300-hPa temperature. Track density in shaded and solid line for contour interval of 4.0 units. Analysis is performed using the ERAI reanalysis for a 36-yr period (1979-2014). Unit is number per season per unit area, the unit area is equivalent to a 5° spherical cap ($\cong 10^6$ km$^2$).

Another point is that using a cold core condition leads to uncertainties between studies since the cold core search is generally performed at different layers, which seems to be chosen arbitrarily in different studies.

We believe that simpler schemes based only on winds should be more representative of reality since they simply impose on the detection the presence of a cyclonic circulation regardless of the physical and dynamical characteristics. The impact of multiple criteria schemes is discussed in this paper:

Pinheiro, H. R., Hodges, K. I., & Gan, M. A. (2019). Sensitivity of identifying cut-off lows in the Southern Hemisphere using multiple criteria: implications for numbers, seasonality and intensity. Climate Dynamics, 53, 6699-6713 (https://link.springer.com/article/10.1007/s00382-019-04984-x)

• Figure 2: The presentation of these results as well as some of the wording in the explaining paragraphs (ie. L209-211) could be improved to make the point of extension to low-levels without extension to the surface clearer. The "sharp decreases" in regions A and C (L211) are difficult to see.

We agree with reviewer. However, we removed this analysis from the paper because it was outside the scope after modifications.

• Figure 3: It may be useful to plot some proxy for the jets on this figure as this is a large part of the authors argument for why deep COLs preferentially occur in specific regions. Does the seasonality of these COL depths coincide with when the split jet occurs (during the cool season)? This discussion should also be expanded.

We acknowledge the reviewer's insightful comment and have expanded our discussion in the revised manuscript to encompass the role of subtropical and polar front jets for deepening COLs. We delve into the specific effects of each jet on different COL types. Please refer to Figures 2, 3, and 4 and the accompanying discussions for further details.

• L243: "Figure 1c" -> "Figure 3c"?

The figure caption has been corrected for accuracy and the figure itself has been moved to the Supplementary Material.

• L242-243: "southeastern Pacific, where deep COLs observed at more northern latitudes" – there doesn't seem to be that much change in latitude from Figure 3. Consider some latitude statistics to prove this point.

This has been removed from the text.

• Figure 4 and Figure 5 - do both of the timesteps provided represent the relevant phases that the author suggests in L254-255? For example shallow COLs at T0 seem to be similar (at least in the upper-levels) to deep COLs at T-48? Do shallow COLs actually ever reach maturity?

The authors are not sure if they understand the reviewer's comment. Composites are produced for particular fields offset from the time of maximum intensity of each track (maturity stage), though it can be defined as maximum growth. Note, however, that the lifecycle stages of deep COLs are more distinguishable due to their stronger gradients and longer lifetimes compared to shallower COLs.

PS: Figures 4 and 5 have been renumbered to Figures 5 and 6, respectively.

• Figure 6: Deep cut-offs appear to be embedded somewhat in really strong westerlies? Is this true? And does this have an impact on the associated baroclinicity? This point is very briefly mentioned (L276), but could be expanded on.

Yes, deep COLs are often embedded in strong westerlies, as demonstrated throughout the study. This is because strong westerlies provide a source of baroclinicity and a favourable environment for their deepening.

The revised manuscript discusses the vertical baroclinic structure in deep COLs and the interdependence between upper- and lower-tropospheric systems due to feedback mechanisms. Specifically, we discuss how the dynamics of deep COLs differs from that of shallow COLs, and how this difference can affect the vertical coupling.

---

## Author Comment (AC2)

**Authors' response to reviewers' comments**

Manuscript Number: egusphere-2023-1996

Exploring the vertical extent and deepening mechanisms of cut-off lows in the Southern Hemisphere: insights from eddy kinetic energy analysis

New suggested title:

**"Deepening mechanisms of cut-off lows in the Southern Hemisphere and the role of jet streams: insights from eddy kinetic energy analysis"**

The authors gratefully acknowledge the insightful feedback from the anonymous reviewers and the editor, which significantly strengthened the revised manuscript. We have carefully considered each comment and implemented substantial changes to address them, particularly regarding the influence of jet streams on COLs. We believe the revised manuscript is now significantly stronger and more impactful.

We have made adjustments by removing certain figures (or relocating them to the Supplementary Material) and shortening the text to prevent excessive lengthening of the paper. Furthermore, we have replaced the term "mid-latitude jet" with "poleward jet" during the discussion, as it is a more precise description.

In this document, the responses to reviewers' comments are highlighted in red font.

**Reviewer 2**

**General comment**

The study considered COLs in the SH and then categorized them according to how deep they. These categories were shallow, medium and deep COLs and the latter were shown to extend to the surface. The study further undertakes an energetics analysis to gain insights into the mechanism that could be responsible for the extension of COLs to the surface. This is a well written and succinct paper. It has the potential of making a contribution to the current work that is ongoing on COLs in the SH. I recommend that it be considered for publication, provided the comments below are adequately addressed.

**Comments**

> **2.** Eq 1. This study aims to consider the three dimensional structure of COLs, so why are the authors then taking the volume integral? This will average our processes that I believe are important to consider if the vertical structure is to be assessed. So, I challenge the authors to consider the EKE that is not integrated to reveal vertical processes (as will be mentioned again later in this review). So I am suggesting that consider the diagnostics used here carefully.
>
> We calculate energetics for both vertically integrated quantities and each pressure level separately. This approach enables us to analyze the EKE budget in a vertical cross-section, as illustrated in Figures 5 and 6. This information can be found in Section 2.3.

**3.** Top panels of Fig 4 and 5. How were the vertical profiles produced? Did you average over a range of latitudes. Please include the jet in this plot, they are, in my opinion, an important piece of the puzzle.

Thank you for pointing out this gap in our study. We produced west-east vertical cross-sections using a system-centered compositing method. A 25° latitude-longitude grid is initially set up on the equator and then rotated to the 300-hPa COL center. The data is then interpolated to this grid at each level from 1000 hPa to 100 hPa. These cross-sections are created at multiple time steps during each COL lifecycle, relative to the time of maximum 300-hPa vorticity intensity. We have clarified the steps involved to produce composites in in Section 2.3.

**4.** Paragraph 270: I agree with the notion that the eddy driven jet is a primary source of EKE and please use the ridge axis as a reference to differentiate between the energy centre that develops from baroclinic conversion and the one that arises as a result of ageostrophic flux convergence. Recent studies in the SH (even though they have focused on regional issues) have shown that there is very weak or no baroclinic conversion downstream of this ridge axis. The authors should explain the cause for the vertical circulation that leads to baroclinic conversion. This is particularly important in this paper because it looks like the authors argue that the midlatitude jet streak is responsible for differences between shallow and deep COLs, and as a reviewer, I totally agree with that.

Thank you for your thoughtful comments. Firstly, we agree that the ridge axis is a useful reference for differentiating between the two energy centers, and we have added this to our discussion.

The reviewer is correct that mid-latitude disturbances associated with cyclones typically exhibit weak baroclinic conversion on the upstream ridge, as shown in the studies by Chang (2000, MWR) and Danielson et al. (2006, Atmosphere-Ocean). There are, however, some important differences between the characteristics described above and the mechanisms that act on COLs, which are predominantly influenced by stationary Rossby waves.

As these Rossby waves break, the trough-ridge system deepens and induces anticyclonic barotropic shear and potential vorticity overturning, as suggested by the reviewer in the following comment. This, in turn, triggers stronger ageostrophic fluxes and momentum from the upstream midlatitude jet into the COL system. Additionally, localized baroclinic processes become prominent on the western flank of COLs, primarily due to descent of cold air on the upstream ridge, enhanced by radiative processes, as outlined in our manuscript. This phenomenon is a robust feature of COL systems, seemingly irrespective of their vertical extent, though stronger ageostrophic fluxes can strengthen the COL and make it more persistent. Further discussion on this topic have been incorporated into the revised manuscript.

**5.** Paragraph 275: I think that question of why the jet is stronger for deeper COLs must be addressed.

Still on this issue: The authors mention and show that the ageostrophic fluxes are stronger for deeper COLs. This makes perfect sense but why? Would the authors agree with the hypothesis that a stronger jet causes increased anticyclonic barotropic shear, which in turn causes higher strain rates and therefore the higher likelihood for RWB (Nakamura and Plumb 1994)? This means that the flow across the ridge axis becomes more super geostrophic, implying a stronger ageostrophic flow across the ridge. But the breaking increases the intensity of the geopotential height anomalies. Thus, combining these two issues leads to stronger ageostrophic fluxes and stronger EKE in the COLs.

We agree with the hypothesis proposed, as it aligns well with our findings. In addition to the previous comment, we believe that the vertical baroclinic structure in deep COLs is also a manifestation of eddy feedback mechanisms arising from an interdependence between upper- and lower-tropospheric eddies. This discussion has been included in the revised manuscript.

I also curious about the ageostrophic fluxes associated with the sub-geostrophic flow that the categorization suggested here appears to be revealing. What does that mean for the energy centre at the inflection point immediately downstream of the ridge axis? No mention is made of this and it looks like it is playing some role.

An interesting observation. The subgeostrophic flow at the top of the trough, indicated by the westward flow, seems to be a result of the negative geopotential height anomalies, contrasting with the supergeostrophic flow on the upstream ridge where positive geopotential anomalies exist, thus inducing fluxes oriented north-eastward into the closed circulation region.

While our discussion focuses on certain aspects, there could indeed be additional factors at play that influence the energy dynamics in these systems, which is still to be explored through further investigation. We have included a brief comment on this issue so that it can guide future work.

Why would a stronger jet streak lead to stronger baroclinic conversion: The authors could invoke jet streak dynamical theory to explain this. For instance the cross front quasigeostrophic theory of Keyser and Shapiro (1986) could used to explain this. Also the location of the baroclinic conversion relative to the jet should be considered to link the strength of jet and the strength of the conversion. The authors should consider the possibility that the midlatitude jet might be curved, which has implications for vertical motion.

**6.** In point 1 above an issue about the diagnostics used was raised. EKE is not visible at the surface for shallow COLs, that is reasonable. They are however very clearly in existence for deep COLs, even though they are weaker than aloft. So, what causes them? Where does this energy come from? That is why taking the volume integral of the energetics terms is not the appropriate thing to do here. I invite the authors to have a look at Fig 7 in Reviere et al (2015). There is a clear

downward flux of EKE and yet the authors say that this flux is small; yet this seems important in explaining where the surface EKE might be coming from.

In conclusion, I think that the use of energetics has a lot of potential in explaining what causes some COLs to extend to the surface and others not. However I am of the opinion that the energetics framework used here is not adequate and I invite the authors to employ the energy equation derived by Reviere et al (2015), without the volume integration to reveal the vertical structure, in particular attempt to explain where the EKE at the surface is coming from. That is lacking in the manuscript.

My suspicion is that deeper COLs last longer and as the authors have raised dissipation issues here, barotropic conversion must be considered.

Therefore, the first two research questions have been addressed, but the third one has not been adequately addressed and I invite the authors to consider the points made above.

We appreciate the reviewer's valuable comments and suggestions. We agree that the current energetics framework requires further refinement to fully elucidate the mechanisms behind COL deepening. Additionally, we acknowledge the potentially significant role of vertical ageostrophic fluxes in COL deepening. While direct computation of these fluxes remains challenging with current data and resources, requiring adjustments to incorporate them, the revised manuscript recognizes their potential importance and emphasizes the need for future research.

We believe the revised manuscript provides sufficient evidence to adequately address the third research question regarding the mechanisms of COL deepening. The incorporated analyses have significantly enhanced the quality of our research and contribute to a more comprehensive understanding of the dynamics of both shallow and deep COLs.

Regarding the barotropic contribution, the previous study by Pinheiro et al. (2022, QJRMS) examined the mechanisms governing energy changes in COLs. They found that ageostrophic fluxes and baroclinic energy conversion can counteract the substantial damping effects of barotropic energy conversion and friction, particularly in stronger systems. During the decay phase, diabatic processes and dispersive fluxes emerge as the primary contributors to COL dissipation. This is evident in Figure 1, which shows minimal or even no contribution from barotropic conversion to the energy decay of COLs.

[Figure]

Figure 1: Temporal evolution of the main EKE terms for a) deep and b) shallow COLs. The terms are baroclinic conversion (blue line), barotropic convertion (red line), agesotrophic flux convergence (green line), convergence of kinetic energy (black line) and residual (dotted line). Fields are vertically averaged within a 15∘ spherical cap region centred on the COL location. Unit is Joule·s$^{-1}$, scaled by $10^{10}$.

---

## Referee Report (RR1)

Review of "Deepening mechanisms of cut-off lows in the Southern Hemisphere and the role of jet streams: insights from eddy kinetic energy analysis" by Pinheiro et. al.
* * *
Suggested outcome: Major Revisions

Scientific significance: Good
Scientific quality: Good
Presentation quality: Good
* * *
This work creates a climatology of cut-off low depth over the Southern Hemisphere and studies the vertical extent and mechanisms that lead to deep cut-offs from an energetics perspective.
* * *
**General comments:**

This study considers the depth of COLs in the southern hemisphere and, by using an energetics perspective, aims to understand the processes at play. The authors have made some progress since the first iteration of the manuscript. The authors tend to simply display the resulting calculations and, in my view, there is still a lack of explanation and detail as to how the processes unfold and why which would increase the impact of their work. As the aim of this work is to understand these processes, I think the authors should try to expand their analysis to highlight the processes at play.

**Major comments:**

1.  There seems to be a lack of dynamical explanations given to explain the observations highlighted by the calculations done. For example, there is an increase in deep COLs with a strengthening in the polar front jet. Can you explain this using the energetics framework used in this work? There is a hypothesis that eddy feedbacks between the surface and the upper-levels. What does this do and how is related to the development of a deep COL versus a shallow COL from an energetics perspective? More detail needs to be added into the discussion and interpretation of the results, particularly associated to how COLs deepen (or not) in the energetics framework chosen, since this is the focus of this work.
2.  There seems to be a link between depth and intensity of COLs (ie. Deep COLs are generally strong). Yet, the authors state that jets affect depth and intensity of COLs differently (eg. L245). Are the four categories defined in this work interdependent? If so, should they be looked at separately and how why do the different jets impact deepening and intensity differently?

**Specific comments:**

*   L98: "if mean separation" – missing "the"?
*   L110: what are the reasons for the choice of extension level (400hPa for shallow; 800hPa for deep).
*   L112-113: where is the track is intensity metric calculated? Is it the most intense value in the track? Or mean? Be specific.
*   L127-128: Do you just mean monthly means? The sentence is hard to understand.
*   L132: "jet streams exhibit relatively small seasonal variations" – is this true of the subtropical jet in the southern hemisphere? The subtropical jet is barely visible in the mean in summer for example (ie. Fig 3).
*   L145: Missing "."?
*   L162: Do you mean Section 2.3?
*   L166: Deep and strong COLs show similar frequencies and distributions – it may be useful to provide some statistics of the various combinations. Are most strong COLs also deep?
*   L184-185: RWB occurs in regions of weak climatological zonal flow and RWB is also associated with the development of COLs. The authors further state in L193-194) that COLs occur in regions

of weakened westelies. However, in the authors results the find that deep COLs occur most frequently during polar jet stream increases during transitions seasons and mention that this is consistent with previous RWB work. Could you clarify this alignment and elaborate?

- L213 and L218: "anticyclonic vorticity" – do you mean anticyclonic barotropic shear?
- L285: "poleward jet shifts to the east" – I do not really see this occurring in your figures. Could you be more specific? Maybe plotting the jets on some of these timelags may help.
- L290: How do we see momentum transfer from the jet into the COL from your results?
- L293: "stationary nature of Rossby waves" – surely we are talking about transient Rossby waves here and not stationary waves?
- Section 3.5: The residual is presumed here to be mostly connected with diabatic processes. This is of course a reasonable assumption. However, is it possible that some of the residual is generated by frictional processes associated with deep COLs? Do the residuals on different surfaces reveal anything here?

---

## Author Response (AR2)

**Authors' response to reviewers' comments**

Manuscript Number: egusphere-2023-1996

**Deepening mechanisms of cut-off lows in the Southern Hemisphere and the role of jet streams: insights from eddy kinetic energy analysis**

Thank you once again for the valuable feedback from the anonymous reviewer. We have incorporated these suggestions into the revised manuscript to enhance the clarity and robustness of our study. In this document, the responses to reviewer's comments are highlighted in red font.

**Reviewer 1**

**General comments:**

This study considers the depth of COLs in the southern hemisphere and, by using an energetics perspective, aims to understand the processes at play. The authors have made some progress since the first iteration of the manuscript. The authors tend to simply display the resulting calculations and, in my view, there is still a lack of explanation and detail as to how the processes unfold and why which would increase the impact of their work. As the aim of this work is to understand these processes, I think the authors should try to expand their analysis to highlight the processes at play.

We have tried to address this general comment by including more discussion and interpretation of the results.

**Major comments:**

1. There seems to be a lack of dynamical explanations given to explain the observations highlighted by the calculations done. For example, there is an increase in deep COLs with a strengthening in the polar front jet. Can you explain this using the energetics framework used in this work? There is a hypothesis that eddy feedbacks between the surface and the upper-levels. What does this do and how is related to the development of a deep COL versus a shallow COL from an energetics perspective? More detail needs to be added into the discussion and interpretation of the results, particularly associated to how COLs deepen (or not) in the energetics framework chosen, since this is the focus of this work.

Thank you for your valuable comment.

Although our findings show a clear relationship between the strengthening polar front jet and the increase in deep COLs, it is difficult to explain this link within the framework of eddy kinetic energy analysis. However, our analysis on energetics shows that deep COLs are positioned near the equatorward exit of a jet, which seems to facilitate the transfer of eddy kinetic energy from the upstream poleward jet into the COL. This feature is evident in the spatial distribution maps (Fig. 3) as well as in our deep COL composites (Fig. 6). We propose that the intensification of the polar front jet amplifies the convergence of ageostrophic fluxes and enhances baroclinic conversion. This, in turn, leads to heightened vertical motions and consequently the COL deepen into the lower troposphere. This

characteristic is particularly pronounced in deep COLs, setting them apart distinctly from their shallow counterparts. We have explored these points, incorporating additional remarks in the results (Section 3.5) and conclusions.

In addressing the aspect concerning the eddy feedback mechanism between surface and upper levels, we recognize that our current energetic framework may not offer a comprehensive understanding of this interaction. However, we propose exploring avenues to examine deeper into the dynamics of these feedbacks in the future. One could involve investigating the role of vertical ageostrophic fluxes, which likely play a role in the deepening processes within COLs. We also highlighted the importance of diabatic processes for the interaction between mechanisms operating at different levels within deep COL systems and the need for their accurate representation in reanalysis data.

2. There seems to be a link between depth and intensity of COLs (ie. Deep COLs are generally strong). Yet, the authors state that jets affect depth and intensity of COLs differently (eg. L245). Are the four categories defined in this work interdependent? If so, should they be looked at separately and how why do the different jets impact deepening and intensity differently?

Yes, it is possible to examine the four categories individually, and we indeed have conducted distinct analysis for each category. However, we made a choice not to present separate analyses for each category, particularly due to limitations in space and the study's objective on elucidating the deepening mechanisms of COLs. This aim is clearly stated in the main objectives of the study and reinforced in the last paragraph of Section 3.2.

In an additional analysis, we performed a comparison involving the four categories by matching the tracks referred to each category. For example, by matching the strong COLs (50th percentile of strongest systems) against the deep COLs (those that extend to 800 hPa or lower), we found 81% of matched tracks, suggesting a significant correlation between the two categories. However, 19% of deep COLs fall into the category of weak COLs, indicating a level of variance within this classification. Similarly, we found 71% of matches between weak and shallow COLs. Therefore, despite strong and deep COLs (as well as weak and shallow COLs) present similar characteristics, these two categories do not exhibit identical systems due to differences in their classifications and the complex dynamics associated with COLs and their deepening mechanisms. A discussion on these statistics have been included in Section 3.2 and Conclusions.

Another aspect concerning the influence of jets on the depth and intensity of COLs can be seen in the updated Figure 4 which now includes scatter plots showing the relationship between COL intensity and jet intensity. A positive correlation is observed between the intensity of both shallow and deep COLs and the subtropical jet, indicating that the subtropical jet likely contributes to the system intensity. While it is unclear why intensified COLs are associated with strengthened subtropical jet, we propose some potential mechanisms which one involves the induction of cyclonic vorticity on the poleward side of the jet, given in the supplementary Figure S2. This hypothesis is supported by the presence of a small-scale

jet stream observed equatorward of COLs, as noted by Ndarana et al. (2021), which plays a role in both the formation and intensification of COLs. Additionally, factors such as temperature gradients and wind shear along the edge of the subtropical jet may contribute to the intensity of COLs. Nevertheless, further investigations are needed to fully understand these relationships.

Ndarana, T., Rammopo, T. S., Chikoore, H., Barnes, M. A., & Bopape, M. J. (2020). A quasi-geostrophic diagnosis of the zonal flow associated with cut-off lows over South Africa and surrounding oceans. Climate Dynamics, 55, 2631-2644.

[Figure]

**Figure S2**: Zonal mean wind (black contour), relative vorticity (dots), and track density for deep COLs in the Southern Hemisphere for a) Summer (DJF), b) Autumn (MAM), c) Winter (JJA) and d) Spring (SON). Unit is as in Fig. 1 for track density. Zonal winds above 25 m.s$^{-1}$ are plotted for 10 m.s$^{-1}$ contour intervals. Blue (red) dots indicate values negative (positive) below (above) $1.0 \times 10^{-5}$ s$^{-1}$, respectively. All fields are represented at the 300-hPa level.

[Figure]

Figure 4 (main manuscript): Scatter plots indicating the relationships between monthly mean COL intensity and jet intensity for (c) polar front jet and (d) subtropical jet.

**Specific comments:**

• L98: "if mean separation" – missing "the"?

It has been changed

• L110: what are the reasons for the choice of extension level (400hPa for shallow; 800hPa for deep).

The chosen pressure levels (400 hPa for shallow and 800 hPa or lower for deep COLs) effectively capture the contrasting vertical extents of COLs. Shallow COLs are typically situated in the upper troposphere, making 400 hPa suitable for their characterization. Conversely, deeper structures of COLs are better represented by 800 hPa or lower. Notably, each of the three categories (deep, medium, and shallow) accounts for approximately 30% of the total COLs, ensuring a balanced representation across different types in our analysis. We have commented how the vertical depths were chosen in Section 2.3.

• L112-113: where is the track is intensity metric calculated? Is it the most intense value in the track? Or mean? Be specific.

This is calculated based on the maximum intensity observed along each track using the minimum 300-hPa relative vorticity in the Southern Hemisphere. We have added this information for better understanding.

• L127-128: Do you just mean monthly means? The sentence is hard to understand.

Yes, we are essentially referring to monthly means. We calculate the average values for each month separately, considering the varying number of days (28-31) in each month, based on the 6-hourly data, and this is done individually for each month and year. We hope this clears up any confusion.

• L132: "jet streams exhibit relatively small seasonal variations" – is this true of the subtropical jet in the southern hemisphere? The subtropical jet is barely visible in the mean in summer for example (ie. Fig 3).

Indeed, we are specifically referring to the fact that the subtropical jet in the Southern Hemisphere typically exhibits relatively small variations in its position rather than its intensity. We have now revised the sentence to make this point clearer.

• L145: Missing "."?

It has been corrected.

• L162: Do you mean Section 2.3?

Yes, it has been changed. Thank you.

• L166: Deep and strong COLs show similar frequencies and distributions – it may be useful to provide some statistics of the various combinations. Are most strong COLs also deep?

Indeed a good point and somewhat related to the second major comment from the reviewer. To address this, we have appended a paragraph outlining the match percentages between deep and strong COLs, as well as between shallow and weak COLs (please see Section 3.2).

• L184-185: RWB occurs in regions of weak climatological zonal flow and RWB is also associated with the development of COLs. The authors further state in L193-194) that COLs occur in regions of weakened westelies. However, in the authors results the find that deep COLs occur most frequently during polar jet stream increases during transitions seasons and mention that this is consistent with previous RWB work. Could you clarify this alignment and elaborate?

Thank you for your insightful comment. The apparent contraction can be clarified by recognizing the differential impacts of subtropical and polar front jets on COLs, a point that may not have been clearly expressed in the previous version of the manuscript but has been addressed in the revised edition following our modifications.

While RWB and COLs typically occur in regions of weakened zonal flow, our findings suggest that the mechanisms conducive to COL deepening are more likely when a strong polar front jet exists upstream of a COL. This observation is discussed in response to the first major comment and also throughout the paper. Specifically, the presence of a strong polar front jet intensifies ageostrophic fluxes, facilitating the transport of EKE northeastward from the poleward jet to the COL.

Our research suggests that the polar front jet plays a role in creating favorable conditions for COL deepening, while the subtropical jet emerges as a significant mechanism for their intensification. We have updated Figure 4 in the paper to incorporate and discuss the relationship between jet intensity and COL intensity, aiming to provide a clearer understanding of these interactions.

• L213 and L218: "anticyclonic vorticity" – do you mean anticyclonic barotropic shear?

Previous studies have demonstrated that the majority of COLs develop from anticyclonic barotropic shear type, however our emphasis here lies on emphasizing the influence of an intense subtropical jet

which generate anticyclonic vorticity anomalies over its equatorward side and cyclonic vorticity anomalies poleward, as demonstrated in Fig. 2 of supplementary material.

• L285: "poleward jet shifs to the east" – I do not really see this occurring in your figures. Could you be more specific? Maybe plogng the jets on some of these timelags may help.

Figure S3 presents the same variables as those depicted in Figure 5 of the main manuscript, with the addition of the 300-hPa zonal wind mean (indicated by the green line). This shows that the decay of the COL coincides with the poleward jet shifting eastward, thereby discontinuing the supply of energy to the system. We have included this figure as supplementary material and appropriately referenced it in the main manuscript.

[Figure]

**Figure S3**: Temporal evolution of shallow COLs in the Southern Hemisphere relative to the time and space of maximum intensity in $\xi_{300}$. The panels depict: (a) vertical cross-sections of total EKE (shaded) with baroclinic conversion (contour); (b) vertically integrated ageostrophic flux convergence (blue and red contours) with EKE (shaded), geopotential height (orange line), zonal wind mean (green line) and ageostrophic fluxes (vectors) at 300 hPa; (c) vertically integrated baroclinic conversion (red contour) with EKE (shaded), geopotential height (orange line) and ageostrophic fluxes (vectors) at 500 hPa; and (d) EKE, geopotential height (orange line) and ageostrophic fluxes (vectors) at 1000 hPa. Contours represent $0.003 \times 10^{10}$ Joule.s$^{-1}$ for integrated quantities, 50 gpm for geopotential height at 300 and 500 hPa, and 20 gpm for geopotential height at 1000 hPa, while total EKE is indicated by $10^9$ Joule.

• L290: How do we see momentum transfer from the jet into the COL from your results?

In atmospheric motions, the transfer of energy is closely linked to the redistribution of mass and momentum. In the revised version, we have removed the sentence we cited the term "momentum".

• L293: "stationary nature of Rossby waves" – surely we are talking about transient Rossby waves here and not stationary waves?

We appreciate your consideration on this point. While it is true that transient Rossby waves play a significant role in atmospheric dynamics, particularly in driving weather systems, there is also substantial evidence supporting the importance of stationary Rossby waves for the development of COLs. It is important to acknowledge that while some COLs can travel large distances within a westerly wave, many exhibit shorter trajectories, eventually remaining stationary or moving westward. This behavior can be related to quasi-stationary waves which effectively act as blocking mechanisms, influencing the movement and persistence of these weather systems.

In response to the reviewer's suggestion, we have opted to replace the previously cited sentence concerning the "linked to stationary nature of…" with a more detailed discussion of the differences between shallow and deep COLs in terms of energetics.

• Section 3.5: The residual is presumed here to be mostly connected with diabatic processes. This is of course a reasonable assumption. However, is it possible that some of the residual is generated by frictional processes associated with deep COLs? Do the residuals on different surfaces reveal anything here?

We agree with reviewer. Indeed, one factor that may contribute to the residual is the unknown contribution from the friction, which is difficult to assess because this is not computed directly. It is possible that stronger systems have a larger residual contribution from frictional effects. We have added a note to address this consideration.

---

## Author Response (AR3)

*Co-editor decision: Publish subject to minor revisions (review by editor)*

*by Irina Rudeva*

*Thank you for revising the manuscript and adding additional comments on the processes of cut-off low deepening and intensification.*

*I agree with the review that it would be good to understand the eddy feedback between the surface and the upper levels and I hope that this can be done in future work.*

*I have a question on the linear relationship between the STJ strength and shallow COL's intensity (Fig.4d). You mention that that relationship may be due to the strong seasonality in both the STJ and COL's strength (Fig.2c,d). Can you clarify this point by looking at anomalies in jet/COL's strength from climatological values for each month instead of the absolute values? I am looking forward to seeing this result before I accept the manuscript for publication in WCD. Thank you*

Dear Editor

Thank you for your question regarding the relationship between subtropical jet strength and shallow COL intensity. We appreciate your suggestion to explore this further.

We have produced a new plot (which was included as Fig. S3 in supplementary material) to show the relationship between anomalies in subtropical jet strength and shallow/deep COL intensity. However, the linear relationship observed in Figure 4d using absolute values is not as evident when considering anomalies (refer to Fig. S3).

[Figure]

**Figure S3:** Scatter plot indicating the relationship between monthly anomalies of COL intensity and subtropical jet intensity. Deep and shallow COLs are depicted by blue and red colors, respectively. Unit is in m.s$^{-1}$ for jet intensity and $10^{-5}$ s$^{-1}$ for COL intensity (scaled by -1).

One possible explanation, as you pointed out, is that the seasonal differences in the subtropical jet strength is more pronounced than the monthly variations from the climatological means, as

seen in Fig. S4. This suggests that the seasonal cycle might be a stronger influence on these variables than the month-to-month anomalies.

[Figure]

**Figure S4:** Monthly variations of mean intensity (black lines) and mean anomaly intensity (red line) of subtropical jet for the period from 1979 to 2014. Unit is m.s$^{-1}$.

We believe that using the absolute values provides a clearer representation of the relationship between the subtropical jet strength and COL intensity in our study. We have incorporated these new figures into the supplementary material and briefly discussed them in the manuscript (please see Section 3.4).

We appreciate your consideration and understanding in this matter. If you have any further questions or require additional clarification, please do not hesitate to let us know.

We look forward to your response.

Sincerely

Authors

---

## Author Response (AR4)

*Co-editor decision: Publish subject to minor revisions (review by editor)*

*I thank the authors for doing extra analysis on the relationship between the subtropical jet and cutoff low intensities. Based on the new plot (Fig.S3), I'd argue that STJ intensity is unlikely to play a role in the intensity of COLs. Therefore, I recommend replacing Fig4d with Fig3S and modifying the text around l.260 ( "the subtropical jet likely plays a role in influencing the system intensity"). The relationship between the anomalies is even smaller than the relationship with the polar jet intensity for which it is said that 'no significant relationship' is found.*

*Additionally, I am wondering why the intensity of the polar jet was taken as the average wind speed between 50 and 65 deg S. Given that the average position of the polar jet is ~ 52 deg S (see, e.g., Simpson et al 2020 https://agupubs.onlinelibrary.wiley.com/doi/full/10.1029/2020JD032835). Therefore, the poleward intensity of the jet is prioritised by that index. How sensitive is the jet intensity and relationship with COL's intensity to the choice of latitudes?*

*As the title of the paper highlights the role of the jets in COL's development, I believe that these are important questions to address.*

*Kind regards,*

*Irina*

Dear Dr. Irina Rudeva

Thank you again for your feedback.

There seems to be an issue regarding how we are interpreting the data concerning the relationship between the subtropical jet and COL intensity. While we acknowledge a connection, removing the seasonal cycle from the data seems to weaken this relationship. This is likely because removing the seasonal cycle reduces the magnitude of subtropical jet anomalies compared to the absolute values.

Figure S4 suggests that strong or weak jet values do not directly correspond to strong or weak anomalies. This is because the seasonal cycle subtraction depends on the month. For example, 1998 appears to have the strongest anomaly, but not the strongest actual jet value. Similarly, the period between 1988 and 1989 shows a large actual jet value without the largest anomaly value. This observation could mask the relationship which is evident when examining the actual jet strengths, highlighting the limitations of treating the annual cycle as a fixed and unchanging factor.

This paper (https://doi.org/10.1175/JCLI-3256.1) makes a discussion on this issue, emphasizing that the annual cycle itself can exhibit variations from year to year. Consequently, the variability of the seasonal cycle suggests that simply removing it through averaging might not fully capture the dynamic relationship between subtropical jet and COLs.

Although we think this relationship is clear, we have attempted to strengthen the discussion on this point in the manuscript.

[Figure]

**Figure S4:** Monthly variations of mean intensity (black lines) and mean anomaly intensity (red line) of subtropical jet for the period from 1979 to 2014. Unit is m.s$^{-1}$.

Regarding the second comment, we acknowledge that the criterion used for detecting the polar front jet is somewhat simplified, given the temporal and spatial variability in its positioning. However, we intentionally used a wide latitudinal range to encompass this variability, following previous studies (e.g. Bals-Elsholz et al., 2001), as outlined in the methodology of our manuscript.

The challenge in identifying jet streams and their characteristics is partly due to their diverse structure often characterized by fragmentation and meandering which varies over time. Obviously the criterion used here may not capture the finer spatial and temporal scale variations but should be adequate to characterize the large scale circulation patterns expected to affect COL development.

Thank you once again for your valuable input.

Sincerely

Authors

---

## Author Response (AR5)

**Public justification (visible to the public if the article is accepted and published)**:
I thank the authors for their responses to my comments.

Regarding point 1, I think a strong seasonal cycle poses a problem to correlation analysis that is similar to a case when time series have trends - if two timeseries trend, they will always be correlated even if there is no link between them. Therefore, I insist on removing the seasonal cycle to say that the intensity of the STJ impacts the intensity of COLs. The next question is then why COLs' intensity has a strong seasonal cycle, similar to the STJ? I think this is an interesting question that is worth a discussion in the paper. If you still want to propose that the seasonality in the STJ leads to seasonality in COLs' intensity, then this relationship should be explained.

**Authors' response:**

We have revised the paper and explored various approaches to address the seasonal cycle and its impact on the relationship between the subtropical jet and COL intensity. These approaches include calculating correlations for each month separately and removing the seasonal cycle and any trend using seasonal differencing on the subtropical jet intensity time series. However, we found that removing the seasonal cycle weakens the relationship between COL and jet intensity.

While we hypothesize potential mechanisms for COL intensification, such as induced wind shear along the subtropical jet edge, supported by observations of small-scale jet streams equatorward of COLs (Ndarana et al. 2021), further work is required to understand these relationships and the seasonal cycle. This is acknowledged in the revised manuscript.

We moved the scatter plots regarding the relationship between COL and jet intensities to supplementary material, but we discussed on the uncertainty regarding this relationship. Also, we believe that the uncertainty between COL and jet intensities does not invalidate previous findings regarding the relationship between (subtropical and polar) jet intensity and COL variability. Additionally, we included a reference supporting the idea that weaker subtropical jets facilitate COL development due to weaker eddy-mean flow interactions (Nie et al. 2023) and reinforced the distinct roles of both polar front and subtropical jets on the COL development, as discussed in Muñoz et al. (2020).

References

Muñoz, C., Schultz, D., Vaughan, G.: A midlatitude climatology and interannual variability of 200-and 500-hPa cut-off lows. Journal of Climate, 33, 2201-2222, 2020.

Ndarana, T., Rammopo, T. S., Bopape, M. J., Reason, C. J., and Chikoore, H.: Downstream development during South African cut-off low pressure systems, Atmospheric Research, 249, 105315, doi:10.1016/j.atmosres.2020.105315, 2021.

Nie, Y., Wu, J., Zuo, J., Ren, H. L., Scaife, A. A., Dunstone, N., & Hardiman, S. C.: Subseasonal prediction of early-summer Northeast Asian cut-off lows by BCC-CSM2-HR and GloSea5. Advances in Atmospheric Sciences, 40, 2127-2134, 2023.

To point 2, I appreciate the complexity and ambiguity of polar jet identification, but I am still concerned about why the wind speed is averaged on the poleward side of the polar jet located at 52S. To my knowledge, Bals-Elsholz et al. (2001) explored the split jet regime around New Zealand, where the polar jet is shifted poleward, however, in other regions its location would be closer to 50S (Simpson et al 2020). If you believe that 50-65S is the true location of the polar jet, it should be better justified (by analysis).

**Authors' response:**

We appreciate the editor's point regarding the need to address the zonal variation in the polar front jet. As shown in the figure below, the seasonally averaged zonal wind at 300 hPa reveals a clear shift in the polar front jet core's latitude. While it is located around 45-50°S across the South Atlantic and South Indian Oceans, it migrates poleward to around 60°S in the Australian region. To address this important aspect, we have included a sentence in the revised manuscript that justifies our chosen latitudinal range for the analysis. Thank you for helping us improve the methodological clarity of our work.

[Figure]

Figure: 300-hPa zonal mean wind in the Southern Hemisphere for Autumn (MAM), Winter (JJA), Spring (SON) and Summer (DJF). Zonal winds are plotted above 25 m.s$^{-1}$ for 5 m.s$^{-1}$ contour intervals.

---

## Author Response (AR6)

**Public justification (visible to the public if the article is accepted and published)**:
I thank the authors for their revision of the manuscript.

I think the relationship in the remaining Fig.4 between the intensity of the jet and the number of COLs comes primarily from the location of the jets as well as identification methods for COLs and jets rather than the jets' intensity. In particular, the 'intensity' of the STJ describes its presence or absence in the first place: a 'weak' STJ is found in summer when there is no defined jet between 25-35degS; whereas a 'strong' STJ stands for winter. Again, should the jet's intensity be replaced with anomalies in Fig.4, I reckon the relationship would be lost. In summer, there are many more shallow COLs on the equatorward side of the polar front jet than in winter, when the presence of the STJ reduces the number of shallow COLs in some parts of the southern hemisphere (e.g., there will be fewer persistent vortices north of the 40S due the presence of the STJ). On the other hand, deeper COLs in winter may be explained by stronger du/dy on the poleward side of STJ contributing to stronger (more cyclonic) relative vorticity, i.e., stronger COL's intensity.

I welcome authors to comment on this relationship. I notice that the number of COLs, especially shallow COLs, also has a strong seasonal cycle, therefore, it needs to be removed similar to the season cyclone in COL's intensity.

**Authors' response:**

Dear Editor

While it is true that the relationship between the COLs and jet intensity in Fig.4 may be influenced by factors such as the location of the jets and the methods used to identify COLs and jets, it is important to consider the broader context. The intensity of the subtropical and polar jets influences the distribution and characteristics of COLs throughout the seasons.

However, replacing jet intensity with anomalies in Fig.4 overlooks this relationship, potentially obscuring important insights. Our findings are supported by recent research, particularly a study authored by the editor, which demonstrates the presence of low-pressure anomalies associated with COLs during 'wet' years in Australia. These anomalies coincide with periods characterized by a stronger polar jet and amplified zonal-wave-3, which corroborates our findings.

http://www.bom.gov.au/research/publications/researchreports/BRR-053.pdf

We maintain confidence in the validity of the connection between COLs and jet intensity. We are somewhat unclear about what we are being asked to do, is anything further required to be modified in the paper or is a response to the editors comment enough. We could add a further comment in the paper concerning the uncertainties in the COL/jet relationship in view of the de-trending along the lines of our response above but it would be useful if we had specific guidance on what is required for the paper to be accepted.

Thank you for your understanding.

Authors

---

## Author Response (AR7)

Dear editor,

We have addressed your suggestions, particularly focusing on the uncertainties between jet intensity and COLs (number and intensity), recognizing the limitations in these associations and methods employed, which still stand as hypotheses. We have emphasized the need for further investigation to clarify these relationships throughout the manuscript. The response to each of your comments are outlined below.

My main concern is that the relationship shown in Fig.4 of the manuscript might result from the seasonal cycle in the strength of the subtropical jet and the number of shallow COLs (Fig2a,d) that will lead to a spurious correlation between them. Please note that I am not saying that there is no relationship, but such a statement should be better supported. This is important to warrant the high quality of manuscripts published in WCD.

Therefore, I request the following changes to the current version of the manuscript:

(1) replace absolute values in Fig 4 with anomalies;

R. We have addressed your suggestion by replacing Figure 4 with a four-panel figure that includes both raw and anomaly values for easier comparison. Additionally, we have kept the relationship between COL intensity and jet intensity to a supplementary Fig. S3.

(2) Modify the text, especially around lines 255-260, to avoid statements that result from what is likely a spurious relationship, e.g. "a significant negative correlation ... suggests a clear relationship between the intensified subtropical jet and the reduction in shallow COLs". The relationship between the intensity of the STJ and the frequency and/or intensity of COLs should remain a hypothesis throughout the text unless solid proof is found.

R. The text has been revised to acknowledge that the initial correlation between jet intensity and shallow COLs observed in raw counts weakens considerably when the seasonal cycle is removed. We have rephrased the discussion to emphasize that this suggests a potential link, but further investigation is needed to confirm a causal relationship. This emphasis has been carried through to the conclusions.

Furthermore, the statement that 'weaker subtropical jets correspond to weaker eddy-mean flow interactions, thereby facilitating the development of COLs ' seems in contrast with the statement by Nie et al. who said that 'underestimated simulated strength of both the Eurasian midlatitude and East Asian subtropical jets may lead to the weaker local eddy-mean flow interaction responsible for the cut-off low evolution.' I understand that weaker jets lead to weaker COLs and vice versa. This agrees with Fig.S3 but not with Fig. 4b, as the number of shallow COLs drops dramatically with stronger STJ but deep COLs do not increase in frequency. So, even if a weaker jet facilitated the development of COLs, then why did not the number of deep COLs rise with the increase of the STJ wind speed? Please edit this sentence to bring fig 4 and S3 in line with Nie et al.

R. To prevent confusion, we have decided to remove the sentence referencing Nie et al.'s statement, as it appears there might be a misunderstanding in how we are interpreting these relationships.

Finally, on the point made in your last response, extreme wet events have been associated with COLs in many studies (e.g., Risbey et al. 2009, Barnes et al.2023). However, the statement by Rudeva et al. reads: 'the extratropics during 'wet' years are characterised by stronger polar jet and amplified zonal-wave 3'; this does not support the idea that the STJ intensity is related to the COLs intensity (please also note that Rudeva et al is not a peer-reviewed publication).

We have focused on the existing literature in the revised version.